# In vivo topology converts competition for cell-matrix adhesion into directional migration

Fernanda Bajanca[1], Nadège Gouignard[1], Charlotte Colle[2], Maddy Parsons [3], Roberto Mayor[2] &
Eric Theveneau [1,2]

When migrating in vivo, cells are exposed to numerous conflicting signals: chemokines, repellents, extracellular matrix, growth factors. The roles of several of these molecules have been studied individually in vitro or in vivo, but we have yet to understand how cells integrate them. To start addressing this question, we used the cephalic neural crest as a model system and looked at the roles of its best examples of positive and negative signals: stromal-cell derived factor 1 (Sdf1/Cxcl12) and class3-Semaphorins. Here we show that Sdf1 and Sema3A antagonistically control cell-matrix adhesion via opposite effects on Rac1 activity at the single cell level. Directional migration at the population level emerges as a result of global Semaphorin-dependent confinement and broad activation of adhesion by Sdf1 in the context of a biased Fibronectin distribution. These results indicate that uneven in vivo topology renders the need for precise distribution of secreted signals mostly dispensable.

[1] Centre de Biologie du Développement (CBD), Centre de Biologie Intégrative (CBI), Université de Toulouse, CNRS, UPS, 118 route de Narbonne, 31062 Toulouse, Cedex 09, France. [2] Department of Cell and Developmental Biology, University College London, Gower Street, London WC1E 6BT, UK. [3] Kings College London, Randall Centre for Cell and Molecular Biophysics Room 3.22B, New Hunts House, Guys Campus, London SE1 1UL, UK. Correspondence and requests for materials should be addressed to E.T. (email: eric.theveneau@univ-tlse3.fr)

Control of directional migration is critical for embryo development and immunity and is often impaired in cancer and chronic inflammation. The composition, organisation and stiffness of the extracellular matrix, secreted factors and cell–cell communication influence directional migration[1–3]. Yet, we poorly understand how cells integrate various, and somewhat conflicting, inputs. There is still much speculation regarding the in vivo function of proposed attractants. Gradients have been observed in vivo, as in the drosophila egg chamber[4], but their existence and relevance in larger structures remains controversial. During migration of the fish lateral line, the distribution of the chemokine Sdf1/Cxcl12 is homogeneous[5]. It is only through differential endocytosis that a gradient of Sdf1 emerges[6,7]. That gradient is the opposite of archetypical hypothesised gradients. It is short-range, steep and transient. The tail of the fish at the time of lateral line migration is a large structure to cross from end to end, in this context robustness may be better achieved with a self-generated gradient rather than a pre-established one. Yet, the tail of the fish embryo at this stage of development is relatively stable in size and shape. There are more complex situations. In the chick embryo, cephalic neural crest cells, a population responsible for most of the peripheral nervous system and craniofacial features of vertebrates[8], undertake migration when the head is dramatically changing. In 24 h, it roughly doubles in length and width[9]. Generating a long-range, stable, shallow gradient in 3D over time under these conditions would certainly be costly. Even more so, if such high maintenance has to be done for multiple molecules. In the cephalic region alone, migrating NC cells are exposed to Eph/ephrins, slit/robo, Semaphorins, VEGFA, PDGFA, FGF8, Sdf1, Fibronectin, Laminins, Collagens and Versicans among others[10–17].

It is currently accepted that (i) Eph-ephrins assign NC cells to subpopulations, that (ii) NC cells invade inhibitor-free corridors of extracellular matrix and (iii) along which they are guided to their final location by attractants such as Sdf1 or VEGFA[10,18]. Yet, Sdf1 in Xenopus and VEGFA in chick embryos are not restricted to target tissues but expressed all along the migratory path[19–22]. Interestingly, directional migration of Xenopus NC cells can be achieved in vitro and in silico solely through cell–cell interactions and confinement[11] indicating that chemotaxis is theoretically dispensable. Further, Sdf1 is not able to compensate for a lack of in vivo confinement through downregulation of Versican[11]. Furthermore, Sdf1 gain and loss-of-function led to unexpected results. In absence of Sdf1, migration was abolished[19] suggesting that Sdf1 is required for migration per se and not only for directionality. In the context of inhibitor-free corridors of matrix, one expects an initial dispersion of cells, even if cells would eventually be mis-targeted. Also, an ectopic source of Sdf1 was sufficient to attract cells into Semaphorin-rich regions[19] and similar observations were made using VEGFA in chick[22]. These data suggest that attractants might not simply give directions but could contribute to the definition of what is a permissive environment for migration. Altogether, these results raise the question of how cells integrate local signals in order to initiate directional migration and what could putative attractants such as Sdf1 or VEGFA do in this context if their distributions are not restricted to target tissues.

To address this question, we used the Xenopus cephalic NC cells as a model and focused on the most-studied positive and negative signals regulating NC migration: sdf1 and class3-Semaphorins[23]. Here we show that Sema3A reduces cell-matrix adhesion, protrusive activity, cell spreading and cell speed and that all these effects are rescued by Sdf1. Sema3A and Sdf1 have opposite effects on Rac1. Direct activation of Rac1 or integrins mimics the effect of Sdf1. Importantly, global activation of cell-matrix adhesion or Rac1 in vivo is sufficient to rescue directional migration in absence of Sdf1. Altogether, our results indicate that in the context of a non-homogenous environment (physical constraints, biased distribution of matrix), a direct competition between pro and anti-adhesion signals at the single-cell level can be efficiently translated into directional migration at the population level. This strongly suggests that in environments with a clear topology, the structuration of putative attractants in large scale gradients is likely to be dispensable.

## Results

**NC cells are surrounded by semaphorins prior to migration.** We first assessed the distribution of Sdf1, Semaphorin 3A and 3 F mRNAs by in situ hybridisation, before migration (Fig. 1a, st17) and throughout migration (Fig. 1a, St21-St28, dorsal views on Supplementary Fig. 1). NC cells are initially lined on their ventro-lateral side by Sdf1 and completely surrounded by Sema3A/3F. In addition, Sema3A, and to a lesser extent Sema3F, is found in the brain dorsally to the NC territory. Discrete distribution of inhibitors and attractants are only observed at late stages of migration (Fig. 1, st23–28). To better appreciate the distribution of Sdf1, Sema3A and 3F with respect to NC cells, we converted images shown in Fig. 1a to false colours, aligned them using morphological landmarks and overlaid them (Supplementary Fig. 1). Overall, our data indicate that premigratory NC cells do not face a pre-patterned environment with inhibitor-free corridors and a chemoattractant at a distance. Instead, NC cells are surrounded by Semaphorins and Sdf1 overlaps with Sema3A/3F on the ventro-lateral side of the NC territory (Fig. 1b, c). Sema3A/3F and Sdf1 are secreted molecules, their area of influence is likely broader than the area of mRNA expression. At later stages, when NC cells are organised in streams, Sema3A marks the anterior and posterior limits of the NC domain whereas Sema3F is expressed dorsally and in between NC streams together with Sdf1. Strikingly, on transversal sections, early migrating crest cells can be seen overlapping with Sema-positive ectoderm (Supplementary Fig. 1), indicating that, at early stages of migration, NC cells do not distribute according to Sema+/Sema− boundaries. This suggests that, at the onset of migration, either cells do not respond to Semaphorins or that class3-Semaphorins are not used to restrict NC migration in Xenopus.

**Knockdown of Sema3A/3F leads to NC migration defects in vivo.** Neuropilin 1 and 2 are expressed in cephalic Xenopus NC cells[24]. Nrp1 and Nrp2 can both act as co-receptors for either Sema3A and/or 3F depending on the cell type studied[25–27]. Therefore, all NC cells could theoretically respond to both Sema3A/3F. In addition, in chick, fish and mouse embryos, class3-Semaphorins are known to restrict NC migration (see ref. [23], and references therein). Thus, NC migration underneath the Sema-positive ectoderm is unlikely to be due to a lack of response from the cells. To assess whether the negative function of semaphorins is conserved in Xenopus, we knocked down Sema3A and 3F using antisense Morpholinos (Fig. 1d, e). On control sides, there are three streams of migratory NC cells (numbered 1, 2 and 3 from anterior to posterior). The first one reaches underneath the eye (Fig. 1d). In absence of Sema3A and/or 3F migration still occurred but streams were shorter (Fig. 1d arrows and black arrowheads) and less defined than controls, an expected effect of lateral dispersion due to lower confinement[11]. Many cells accumulated dorsally (Fig. 1d, red arrowheads) or in between streams. Most embryos showed asymmetrical distribution of NC cells when comparing control and injected sides (Fig. 1h). Around 70% of all embryos with Sema3A and/or 3F MO had NC cells in ectopic locations: over the eyes, in between

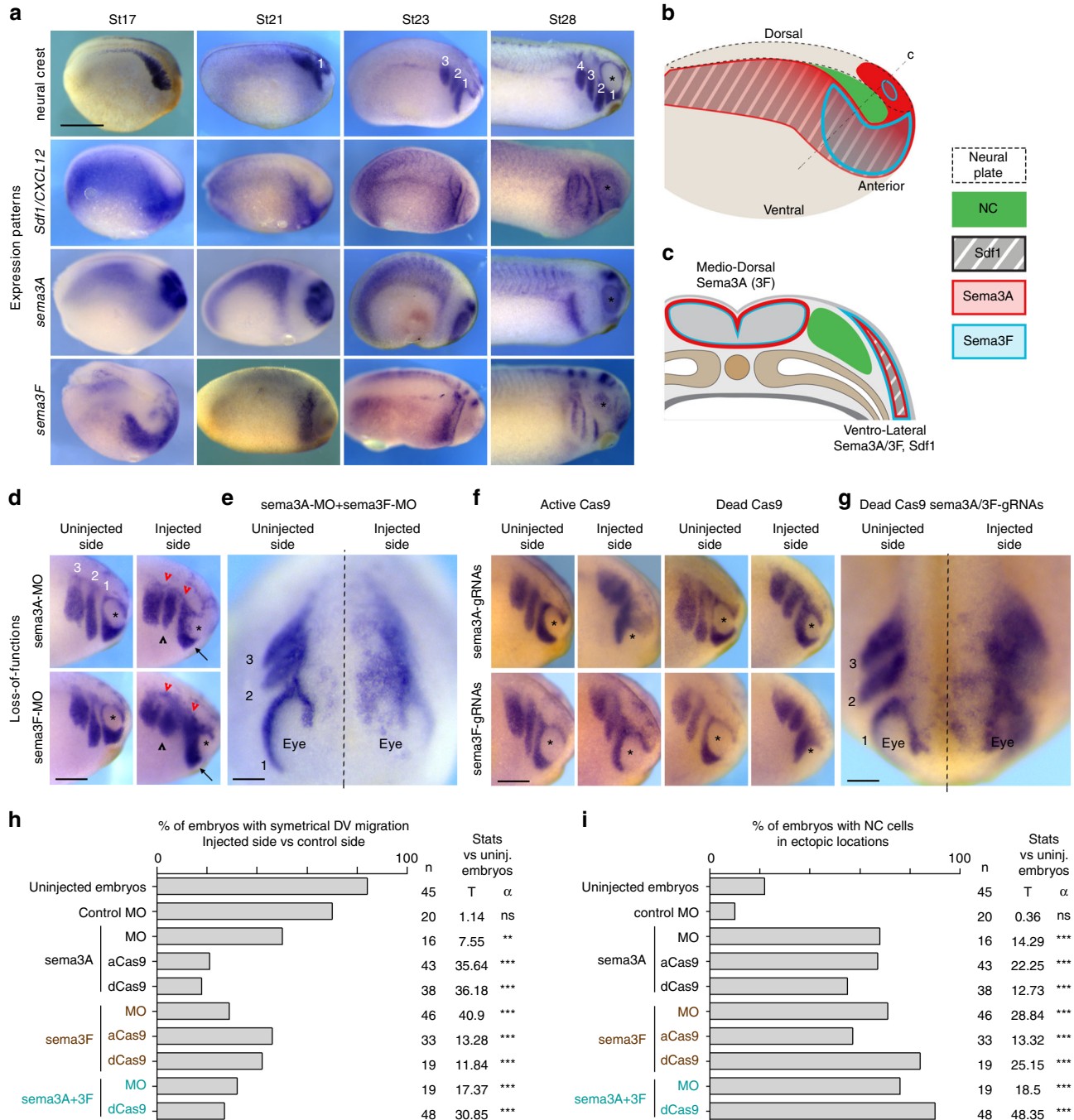

streams, between the NC domain and the dorsal midline (Fig. 1i). To substantiate these data, we performed *sema3A/3F* knockdowns using CRISPR/Cas9 by co-injecting either an active Cas9 (aCas9) or a transcription-blocking catalytically inactive mutant form (dead Cas9, dCas9) with a cocktail of three guide RNAs against *sema3A* and/or *3F* (Fig. 1f, g)[28]. Efficiency of CRISPR/Cas9 was assessed by in situ hybridisation (Supplementary Fig. 2). Importantly, injecting either active or dead Cas9 without gRNAs did not affect Twist expression or NC migration (Supplementary Fig. 2). In addition, injecting the gRNAs against *sema3A* or *3F* without Cas9 did not affect Twist expression or NC migration (Supplementary Fig. 2). These internal controls for CRISPR specificity gave clear results. Thus, we did not further check for genomic modifications. Overall, using CRISPR/Cas9, we observed phenotypes similar to those obtained with the MOs and all treatments

led to significant increases in embryos with asymmetrical left-right NC migration (Fig. 1h) and embryos with NC cells in ectopic locations (Fig. 1i). To further ensure that NC identity was not affected by the CRISPR/Cas9 treatments, we performed in situ hybridisation against Slug and found no downregulation (Supplementary Fig. 2). To obtain a dynamic view of how cells migrate in control and semaphorin knockdown (sema-KD) environments in vivo, we performed time-lapse imaging of NC migration (Fig. 2, Supplementary Movie 1). We grafted fluorescent control NC cells into control embryos or embryos injected with sema3A-MO or sema3A gRNAs with aCas9 or dCas9 (Fig. 2a). Embryos were monitored from 4 h after grafting. When control NC are grafted into control embryos the first stream of cells is already starting to migrate along the eye after 4 h (Fig. 2b, arrowhead) whereas in the sema-KD conditions it has not,

**Fig. 1** *Sema3A* and *3F* are co-expressed with *Sdf1* and restrict NC migration in vivo. **a** In situ hybridisation for neural crest markers (st17, *slug*; st21–28, *twist*), *Sdf1*, *Semaphorin-3A* and *3F*. NC cells migrate as streams numbered 1–4, anterior to posterior. Asterisk marks the eye. **b, c** Diagrams depicting the relative distribution of *Sdf1* (grey), *Sema3A* (red) and *Sema3F* (blue) with respect to NC cells (green) at stage 17 in wholemount (**b**) or transversal section (**c**). **d** Loss-of-function with antisense Morpholinos for *Sema3A* and *3F* analysed by in situ hybridisation for *twist*, embryos st25. Arrows indicate cells from stream 1 that did not reach the area ventral to the eye. Black arrowheads indicate shorter streams. Red arrowheads, cells accumulated in dorsal region. Asterisks mark the eye. Note cells migrating over the eye on the injected sides. **e** Anterior view of a representative embryo injected with both MOs against *Sema3A* and *3F*. Dotted line marks the midline. **f** Loss-of-function with CRISPR/Cas9 and gRNAs for *Sema3A* and *3F* analysed by in situ hybridisation for *twist*, embryos st25. **g** Anterior view of a representative embryo injected with gRNAs against *Sema3A* and *3F* together with dCas9. **h** Percentages of embryos with symmetrical migration along the dorso-ventral axis on non-injected and injected side. **i** Percentages of embryos with NC cells in ectopic locations (over the eye, in between streams, caudal expansion, between midline and NC streams). Proportions for each experimental conditions were compared to control uninjected embryos. 327 embryos were used obtained from three independent experiments, *n* of embryos per conditions are indicated on the figure. Comparisons of proportions were made using contingency tables[60]. Null hypothesis rejected if T > 3.841 (*$\alpha$ = 5%); T > 6.635 (**$\alpha$ = 1%); T > 10.83 (***$\alpha$ = 0.1%). Note that normal NC migration in control embryos displays some level of randomness. Around 15% of non-injected embryos had noticeable differences between their left and right sides. The front of migration was more ventral on one side than the other. Also, about 20% of non-injected embryos had some cells that would be counted as ectopic in experimental embryos, mainly cells located dorsally to the streams, seemingly late. Scale bars are 500 μ, except for **e** and **g**, 50 μ. Source data are provided as a Source Data file

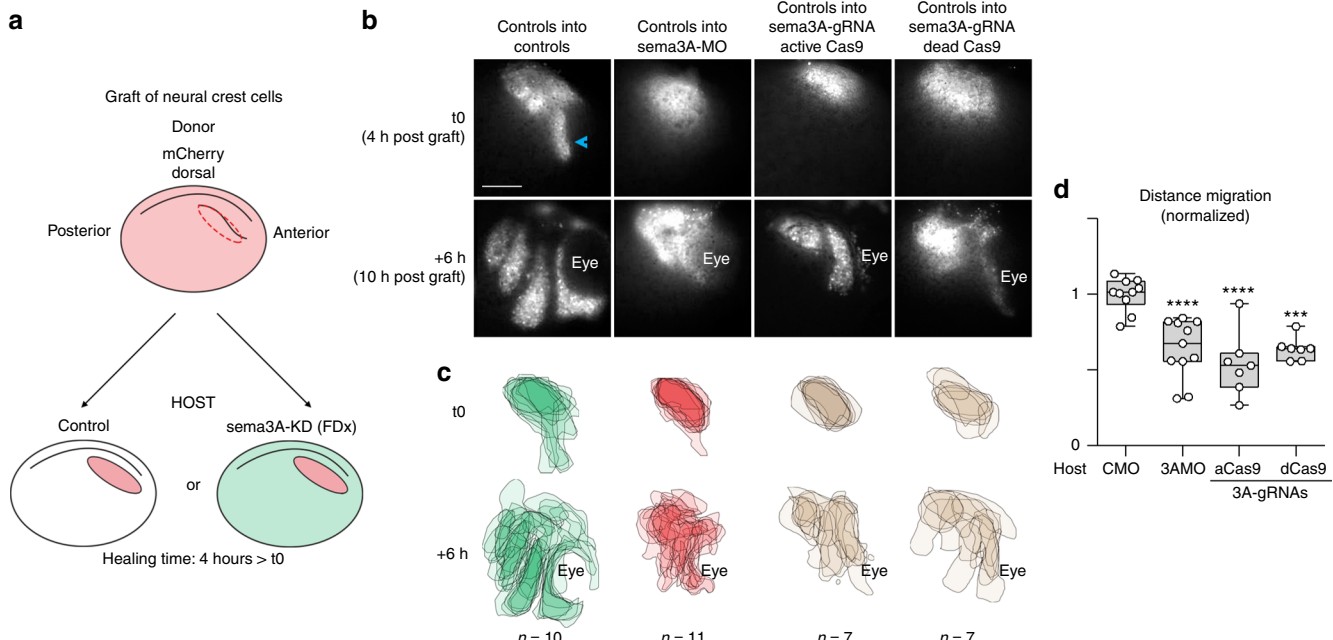

**Fig. 2** Neural Crest migration in control and sema3A-KD environments. **a** Procedure for grafting of NC cells. **b** Images at t0h and t6h of the time lapse movies corresponding to 4 h and 10 h after the graft, respectively. **c** Projections of all NC domains at t0h and t6h for all experimental conditions, note that overall distribution and dorso-ventral migration of control NC cells are affected in sema-KD background. **d** Graph plotting the distance migrated by NC cells in all conditions shown in **b**, **c**. 35 embryos ($n_{CMO}$ = 10, $n_{Sema3AMO}$ = 11, $n_{Sema3A-gRNA-aCas9}$ = 7, $n_{Sema3A-gRNA-dCas9}$ = 7), from five independent experiments. ANOVA followed by multiple comparisons. ****$p < 0.0001$; ***$p < 0.001$. Scale bar 150 μ. Box and whiskers plot: the box extends from the 25th to the 75th percentile; the whiskers show the extent of the whole dataset. The median is plotted as a line inside the box. Source data are provided as a Source Data file

indicating a delay of the onset of migration. After 6 h of time-lapse (10 h post-graft), all NC cells are migrating dorso-ventrally but cells in sema-KD embryos have not reached ventral portions of the face (Fig. 2c) and there is a significant reduction of the total distance migrated (Fig. 2d).

**Sdf1 rescues Sema3A-dependent inhibition of NC dispersion.** Given that semaphorin knockdown gives rise to both ectopic migration and shorter dorso-ventral distances migrated, our data strongly suggest that sema3A/3F act as negative cues constraining NC migration such as Versican[11]. Nonetheless, NC cells seem to ignore Sema3A/3F at the onset of migration and invade directly underneath the Sema-positive ectoderm (Supplementary Fig. 1). Since *Sdf1* expression overlaps with the early ventro-lateral expression of *Sema3A/3F*, we hypothesised that Sdf1 might allow

cells to migrate underneath the Sema-positive ectoderm. To test this idea, we performed a series of in vitro experiments (Fig. 3). We plated NC cells on Fibronectin-coated dishes (Fig. 3a) with or without Sema3A at different concentrations and/or Sdf1 added in the medium (Fig. 3b, c). Cell dispersion was monitored for 8 h. We plotted the whole distribution of each population per hour. Sema3A (grey boxes) has a dose-dependent negative effect on NC cell dispersion and adding Sdf1 (boxes with thick lines) rescues cell dispersion (Fig. 3c). Then, we compared all dataset per time point to identify when a given condition deviates from the control. It took respectively 3, 2 and 1 h for low, mild and high concentrations of Sema3A to significantly reduce dispersion of NC explants (Fig. 3c, grey boxes). Importantly, adding Sdf1 to the medium improved dispersion in all conditions (Fig. 3c, compare boxes with thin (no Sdf1) and thick lines (Sdf1)). For representative examples of explants dynamics for each condition see

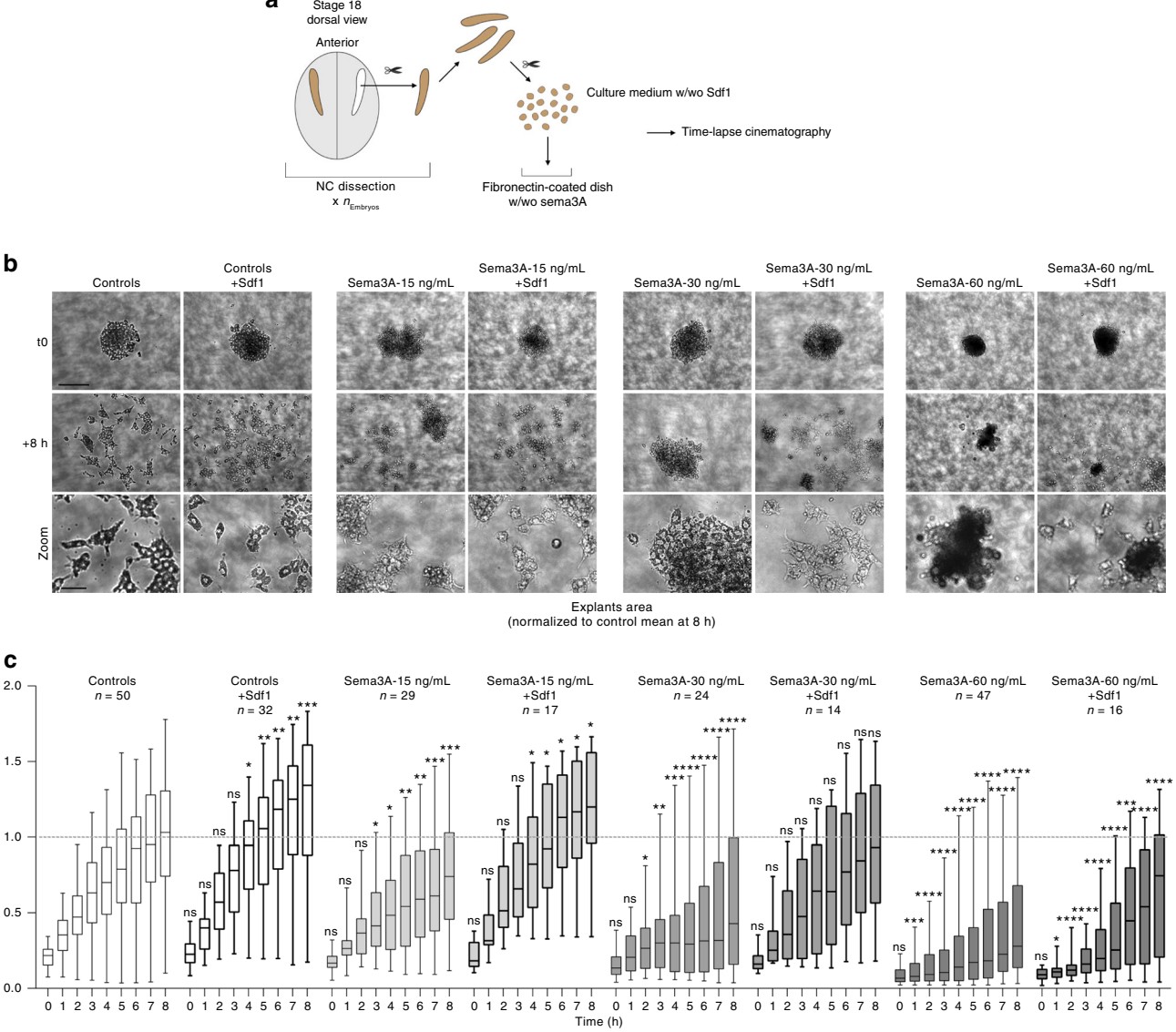

**Fig. 3** Sdf1 and Sema3A have antagonistic effects on cell dispersion. **a** Diagram explaining how NC explants were prepared. **b** Representative examples of explants at t0 (one hour after plating on Fibronectin) and +8 h. Note that cells exposed to Sema3A have a round morphology and tend to stay as small clusters, even when dispersion is rescued by Sdf1. **c** Distribution of explants areas per hour per experimental condition. A total of 229 explants ($n_{controls}$ = 50; $n_{Sdf1}$ = 32; $n_{Sema3A-15ng.mL}$ = 29; $n_{Sema3A-15ng.mL+Sdf1}$ = 17; $n_{Sema3A-30ng.mL}$ = 24; $n_{Sema3A-30ng.mL+Sdf1}$ = 14; $n_{Sema3A-60ng.mL}$ = 47; and $n_{Sema3A-60ng.mL+Sdf1}$ = 16) from five independent experiments were used. Two-way ANOVA, matching: stacked, pairwise multiple comparisons. *$p$ value < 0.05; **$p$ value < 0.01; ***$p$ value < 0.001; ****$p$ value < 0.0001. Dotted line on the graphs represents the mean value for controls at 8 h, provided as a visual reference for comparison with other conditions. Scale bar on low magnification, 100 μ. Scale bar on zooms, 20 μ. Box and whiskers plot: the box extends from the 25th to the 75th percentile; the whiskers show the extent of the whole dataset. The median is plotted as a line inside the box. Source data are provided as a Source Data file

Supplementary Movies 2 and 3. We performed similar experiments with Semaphorin 3F and found that it also has a dose-dependent effect on NC cell dispersion (Supplementary Fig. 3, Supplementary Movie 4). This effect being milder than that of Sema3A, we focused on Sema3A.

**Sdf1 drives NC entry into Sema3A-positive territory.** As migration proceeds in vivo, *Sdf1* and *Sema3A* expressions become restricted to discrete locations and no longer overlap (Fig. 1a, st28). Thus, when provided with Sema-/Sema+boundaries NC cells might preferentially migrate on Sema-free areas regardless of Sdf1. To test this idea, we plated cells on Sema3A and Fibronectin stripes and placed an Sdf1-soaked bead as a source of Sdf1 within

the Sema-positive domain (Supplementary Fig. 4, Supplementary Movies 5–7). NC cells initially respected both signals by migrating towards Sdf1 while staying within the Sema-free corridor. However, NC cells later violated the Sema-/Sema+boundary to migrate towards Sdf1 (Supplementary Fig. 4). These experiments show that migration towards a source of Sdf1 can occur while respecting a semaphorin boundary but that high doses of Sdf1 eventually override semaphorins' negative effect.

**Sdf1 rescues Sema3A's effect on shape and adhesion.** While performing the dispersion assays, we noticed that many explants and single cells exposed to Semaphorins detached from the substrate. To quantify this, we performed a cell adhesion assay

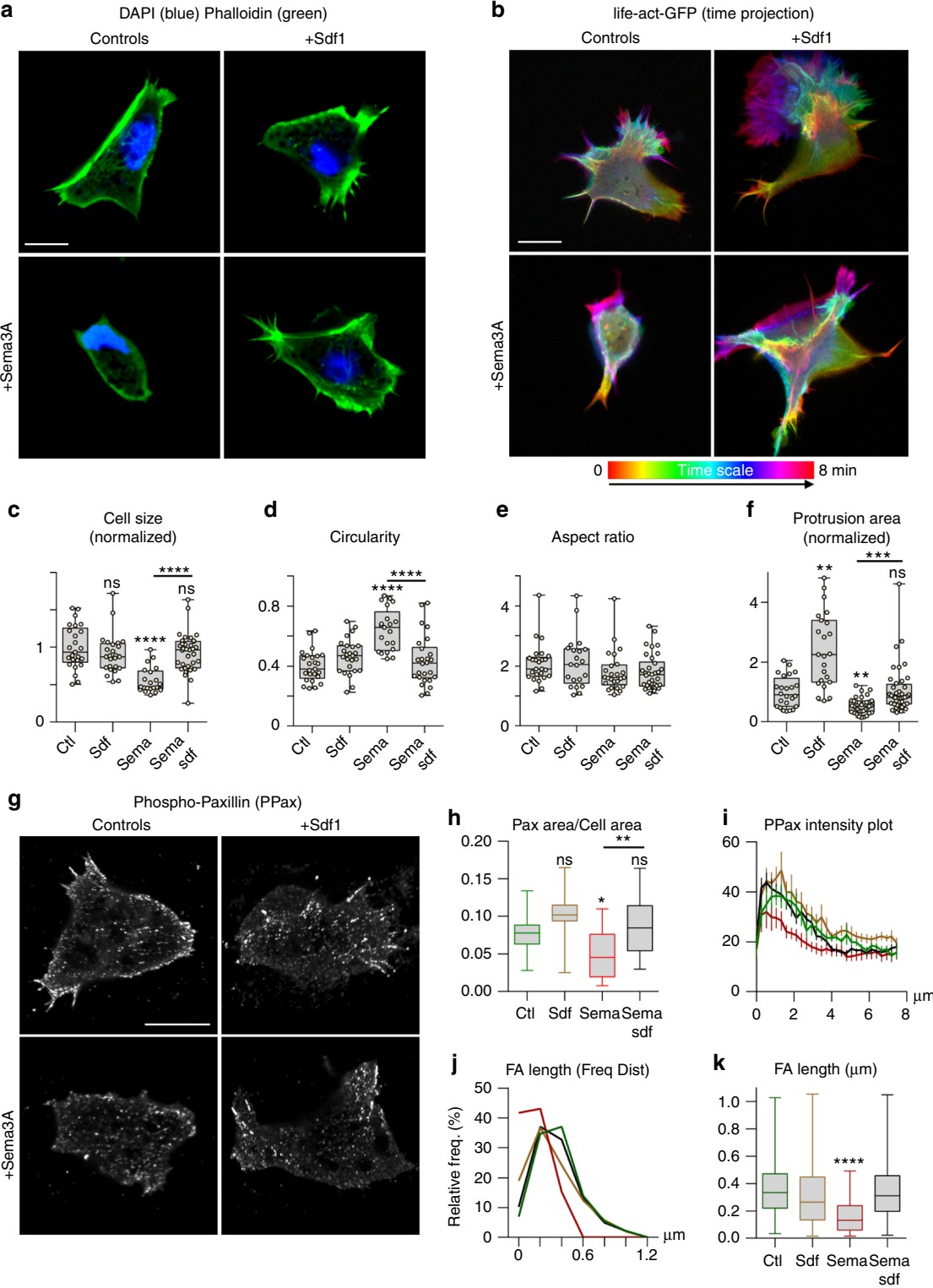

and confirmed that Sema3A impaired adhesion, an effect rescued by Sdf1 (Supplementary Fig. 5). We then looked at cell spreading, circularity, aspect ratio and cell protrusions performing Phalloidin staining (Fig. 4a) and time-lapse imaging using Life-Act reporter (Fig. 4b, Supplementary Movie 8). Sdf1 did not significantly change spreading, circularity or aspect ratio (Fig. 4c–e) but increased the size of cell protrusions (Fig. 4f) as previously reported[19]. Sema3A reduced spreading, circularity and protrusions and these effects were rescued by Sdf1 (Fig. 4c–f). We then

looked at Focal Adhesions (FAs). We first let cells adhere to Fibronectin before adding Sema3A and/or Sdf1 in solution 30 min before fixation. This allowed us to assess direct effect on FAs without bias on cell area. Sdf1 and Sema3A had antagonistic effects on FAs (Fig. 4g), including the total area occupied by FAs (Fig. 4h) and their polarised distribution from the cell tip to the cell's centroid (Fig. 4i). Both effects are due to a loss of large FAs (Fig. 4j, k). Importantly, adding Sdf1 restored all values to control levels. The actin cytoskeleton looked dramatically affected

**Fig. 4** Sdf1 and Sema3A have opposite effects on spreading, adhesions and protrusions. **a** DAPI (blue) and Phalloidin staining (green) on NC cells cultured in control conditions or exposed to Sdf1 and/or Sema3A. **b** Colour-coded time projection of time-lapse movies of cells transfected with life-Act-GFP (see Supplementary Movie 8). **c–e** Area occupied by the cells, normalised to control conditions (**c**), circularity (**d**) and aspect ratio (**e**), $n = 105$ cells ($n_{controls} = 25$; $n_{Sdf1} = 21$; $n_{Sema3A15ng.mL} = 25$; $n_{Sema3A15ng.mL+Sdf1} = 34$) from two independent experiments, from fixed cells counterstained with Phalloidin. **c–e** ANOVA followed by multiple comparisons; ****$p < 0.0001$. **f** Average area occupied per protrusion (from Life-Act movies), normalised to controls, $n = 122$ protrusions ($n_{controls} = 24$; $n_{Sdf1} = 23$; $n_{Sema3A15ng.mL} = 35$; and $n_{Sema3A15ng.mL+Sdf1} = 40$) analysed from two independent experiments. ANOVA followed by multiple comparisons ** (ctl vs sdf), $p = 0.0014$, ** (ctl vs Sema), $p = 0.0048$, ***$p = 0.0009$. **g** Phospho-paxillin (PPax) immunostaining, cells adhering on Fibronectin, Sema3A/Sdf1 were added in solution 30 min before fixation. **h** Ratio of area occupied by PPax divided by area of the cell, $n = 51$ cells from one experiment. Student's $t$-test; *$p = 0.0427$. **$p = 0.0144$. **i** PPax intensity plot from cell tip to cell centroid (mean+s.e.m). Colour code corresponds to conditions shown in graphs **e** and **h**. **j** Frequency distribution of FA length (main axis). Note that exposure to Sema3A reduces the number of large FAs. Colour code corresponds to conditions shown in graphs **e** and **h**. **k** Average FA length, ANOVA followed by multiple comparisons; ****$p$ value <0.0001. Data shown in **h–k** were gathered from two independent experiments, $n = 30$ cells per conditions. Scale bars 10 μ. Box and whiskers plot: the box extends from the 25th to the 75th percentile; the whiskers show the extent of the whole dataset. The median is plotted as a line inside the box. Source data are provided as a Source Data file

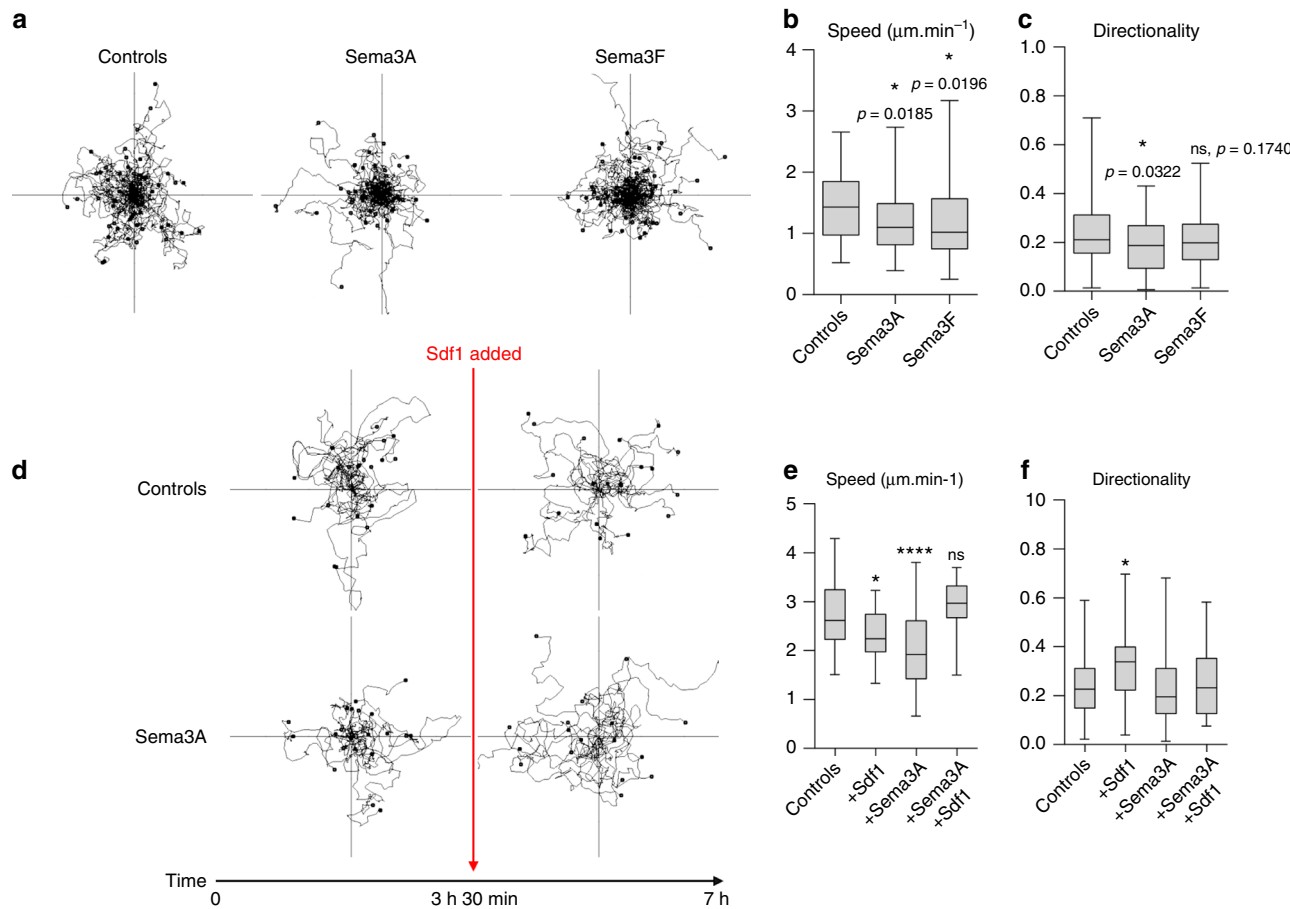

**Fig. 5** Sdf1 counteracts Sema3A's effects on single cell motility. **a** Tracks of single cells on Fibronectin or Fibronectin plus Sema3A. **b**, **c** Speed and Directionality in cells shown in **a**, $n = 188$ cells ($n_{controls} = 64$; $n_{Sema3A} = 53$; $n_{Sema3F} = 71$). ANOVA, followed by multiple comparisons. **d** Tracks of single cells cultured on Fibronectin or Fibronectin plus Sema3A before and after Sdf1 was added in solution. **e** Speed of cells in conditions shown in **d**, ANOVA, followed by multiple comparisons; ns, $p = 0.621$; *$p = 0.0188$; ****$p < 0.0001$. **f** Directionality in cells shown in **d**, ANOVA, followed by multiple comparisons; *$p = 0.0331$. Note that Sdf1 reduces slightly speed and increases directionality whereas Sema3A strongly reduces both. Sdf1 restores control values when added to the Sema3A condition. $n = 190$ cells ($n_{controls} = 60$; $n_{Sdf1} = 29$; $n_{Sema3A} = 73$; $n_{Sema3A+Sdf1} = 28$). Box and whiskers plot: the box extends from the 25th to the 75th percentile; the whiskers show the extent of the whole dataset. The median is plotted as a line inside the box. Source data are provided as a Source Data file

(Fig. 4a, b, Supplementary Movie 8), we thus wondered whether microtubules might be affected as well but found no effect of Sdf1 or Sema3A (Supplementary Fig. 6, Supplementary Movie 9). Finally, we performed cell tracking to assess whether adhesion defects were translated into motility defects. Sema3A/F reduced

motility (Fig. 5a–c) and adding Sdf1 was sufficient to rescue motility (Fig. 5d–f).

**Rac1 activation rescues Sema3A's inhibition of protrusions.** Actin dynamics is regulated by small GTPases[29] and

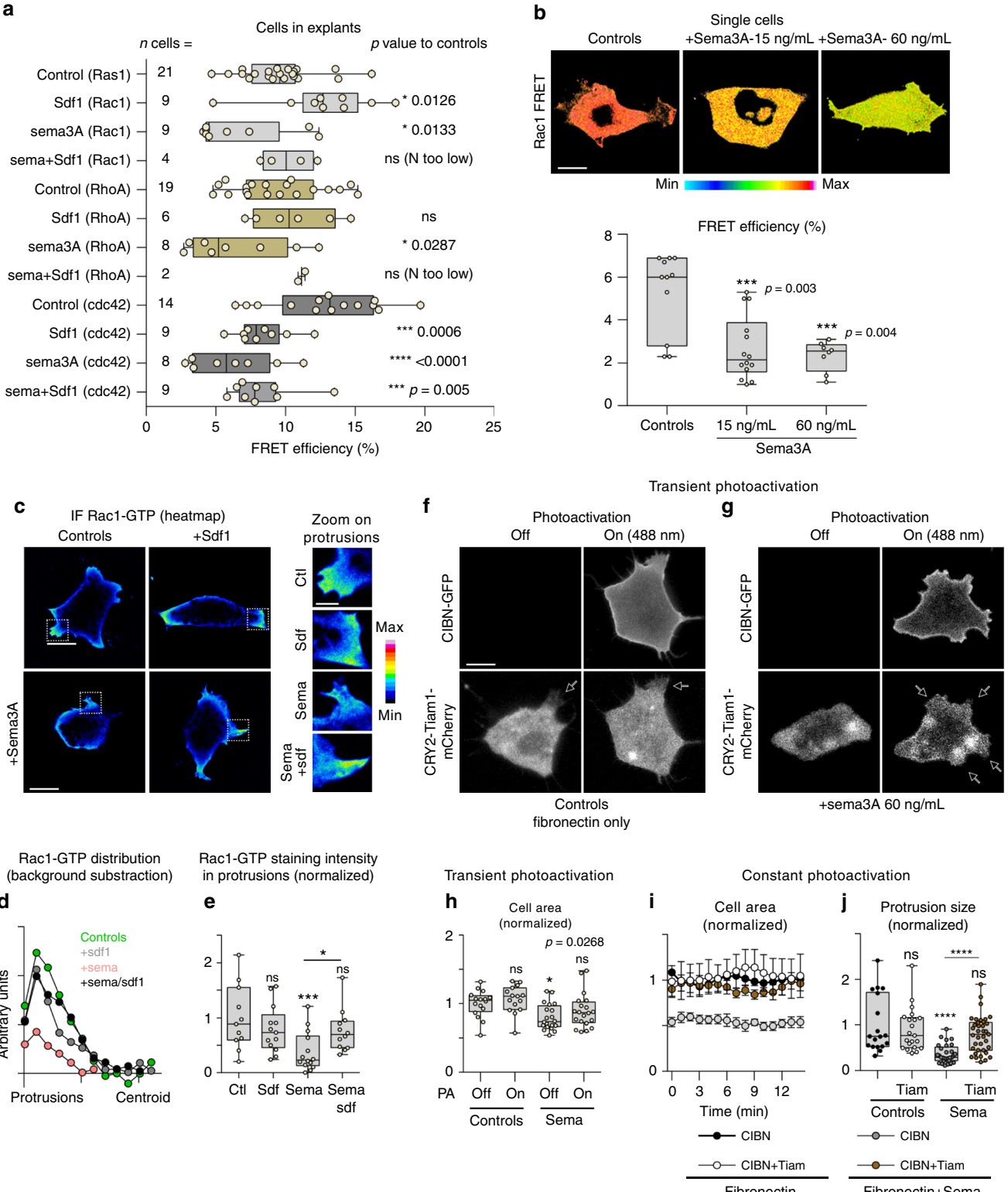

Semaphorins and Sdf1 are able to modulate small GTPases[30–33]. Thus, we assessed the effect of Sema3A and Sdf1 on Rac1, RhoA or Cdc42 activities in NC cells, using FRET reporters (Fig. 6a). Sema3A reduced activities of all three. Sdf1 activated Rac1 (Fig. 6a, light grey boxes), as previously known[19] but had no effect on RhoA (Fig. 6a, yellow boxes) and lowered Cdc42 (Fig. 6a, dark grey boxes). Adding Sdf1 to Sema3A was sufficient to restore seemingly normal Rac1 and RhoA levels but had no

effect on Cdc42 (Fig. 6a). These data indicate that Sdf1 and Sema3A have opposite effects on Rac1 in NC cells in explants. We then performed the FRET assay on cells plated individually (Fig. 6b) to avoid feedbacks from cell–cell adhesion. We confirmed that Sema3A decreased Rac1 activity. To substantiate these data, we assessed Rac1-GTP distribution by immunofluorescence (Fig. 6c). Sema3A reduced polarised distribution of Rac1-GTP (Fig. 6c, d) and intensity in protrusions (Fig. 6c, zooms and 6e).

**Fig. 6** Rac1 activation is sufficient to rescue exposure to Sema3A. **a** Rac1, RhoA and Cdc42 activity assessed by FRET in cells from explants cultured in control, Sdf1 and/or Sema3A conditions, $n = 118$ cells (Rac1: controls(21), Sdf1(9), Sema3A(9), Sema3A + Sdf1(4); RhoA: controls(19), Sdf1(6), Sema3A (8), Sema3A + Sdf1(2); Cdc42: controls(14), Sdf1(9), Sema3A(8), Sema3A + Sdf1(9)) from three independent experiments. For each FRET probe, Sdf1, Sema3A and Sdf1+Sema3A conditions were compared to their cognate controls via ANOVA followed by multiple comparisons, individual $p$ values are indicated on the figure. **b** Rac1 FRET in single cells under control conditions (Fibronectin) or with Fibronectin plus Sema3A coated at 15 or 60 ng/mL, $n = 33$ cells ($n_{controls} = 11$, $n_{Sema3A-15ng.mL} = 14$ and $n_{Sema3A-60ng.mL} = 8$), ANOVA followed by multiple comparisons, $p$ values indicated on the graph. **c** Immunofluorescence against Rac1-GTP. **d** Distribution of rac1 intensity from cell protrusions to cell centroid. Rac1 staining intensity in the cell centroid was measured in each condition and subtracted from each data set, $n = 50$ cells from one experiment. **e** Rac1 staining intensity in protrusions for each experimental condition, $n = 50$ protrusions from one experiment, ANOVA, followed by multiple comparisons; *$p < 0.05$; ***$p < 0.001$. **f** Photoactivation experiment with single cells transfected with CIBN-Caax-GFP and Tiam1-CRY2-mCherry under control conditions. **g** Photoactivation experiments with single cells transfected with CIBN-Caax-GFP and Tiam1-CRY2-mCherry cultured on Fibronectin, plus Sema3A coated at 60 ng/mL. **h** Normalised cell area for experimental conditions displayed in **f** and **g**, $n = 71$ cells ($n_{controls-PA/OFF} = 16$, $n_{controls-PA/ON} = 15$, $n_{Sema3A-PA/OFF} = 20$, and $n_{Sema3A-PA/ON} = 20$) from seven independent experiments, ANOVA followed by multiple comparisons, $p$ value indicated on the graph. **i** Cell area overtime (mean+s.e.m) for cells under sustained photoillumination after being transfected with CIBN and Tiam on Fibronectin or Fibronectin plus Sema3A coated at 60 ng/mL or cells transfected with CIBN only on Fibronectin plus Sema3A coated at 60 ng/mL, $n = 21$ cells from one experiment ($n_{CIBN/FN} = 3$, $n_{CIBN+Tiam/FN} = 4$, $n_{CIBN/Sema} = 8$, and $n_{CIBN+Tiam/Sema} = 6$). **j** Size of protrusions from cells used in **f** and **g**, $n = 105$ protrusions ($n_{CIBN/FN} = 18$, $n_{CIBN+Tiam/FN} = 21$, $n_{CIBN/Sema} = 29$, $n_{CIBN+Tiam/Sema} = 37$). ANOVA followed by multiple comparisons, ****$p < 0.0001$. Scale bars 10 μ, except for zooms in panel **c**, 3 μ. Box and whiskers plot: the box extends from the 25th to the 75th percentile; the whiskers show the extent of the whole dataset. The median is plotted as a line inside the box. Source data are provided as a Source Data file

Sdf1 restored both values to control levels (Fig. 6c–e). Rac1 promotes actin polymerisation and contributes to FA assembly. Therefore, we wondered if activating Rac1 might be sufficient to rescue exposure to Sema3A. We made use of a photoactivatable form of the Rac1 GEF Tiam1[34]. We transfected NC cells with CRY2-Tiam1-mCherry and CIBN-CaaX-GFP. CIBN acts as a docking site for CRY2. The CIBN-CRY2 interaction is controlled by exposure to light under 500 nm and is reversible. In absence of illumination, Tiam1 is cytoplasmic. When exposed to blue light, CRY2 and CIBN bind to one another, recruiting Tiam1 to the cell membrane where it activates endogenous Rac1. This system was tested in mammalian cells[34] and we confirmed that it works in our cells (Supplementary Fig. 7). We then cultured NC cells on Fibronectin with or without Sema3A, performed cycles of illumination with a 488 nm laser and measured the area of cells with or without photoactivation (Fig. 6f, g). Control cells formed protrusions regardless of light exposure (Fig. 6f, arrows) while cells exposed to Sema3A were inactive (Fig. 6g, OFF). When turning on the laser, cells exposed to Sema3A rapidly formed protrusions (Fig. 6g, ON, arrows, Supplementary Movie 10). We then kept cells transfected with or without Tiam1 on Fibronectin or exposed to sema3A under constant photoactivation (Fig. 6I, j, Supplementary Movie 11). Under sustained photoactivation, the average area occupied by control cells and cells expressing Tiam1 in Sema3A conditions were similar whereas cells that were not transfected with Tiam1 cultured under Sema conditions were half smaller (Fig. 6i). Photoactivation of Tiam1 was also sufficient to rescue protrusion size in cells exposed to Sema3A (Fig. 6j). Thus, increasing endogenous Rac1 activity, via Tiam1, is sufficient to promote spreading and protrusive activity in cells exposed to Sema3A.

**$Mn^{2+}$ rescues Sema3A's inhibition of adhesion.** Since Rac1 is upstream of actin polymerisation and FAs, we cannot know if our rescue using Tiam1 is due to an effect on FAs, actin or both. Thus, we made use of Cucurbitacin E (CuE), a microfilament stabiliser[35] and Manganese ($Mn^{2+}$) an activator of integrins including the β1 subunit[36], the main Fibronectin co-receptor in *Xenopus* NC cells[37] (Fig. 7a). FA signalling can feed back into Rac1[38] and we first assessed the effect of $Mn^{2+}$ on Rac1 levels. Exposure to $Mn^{2+}$ significantly increased Rac1 levels (Fig. 7b) indicating that activating integrins with $Mn^{2+}$ might also stabilise actin via Rac1 activation. Then, single cells were plated in control or Sema3A conditions with either $Mn^{2+}$ or CuE (Fig. 7c, d). $Mn^{2+}$ and CuE

did not affect cell spreading under control conditions (Fig. 7d). $Mn^{2+}$ was able to rescue spreading under Sema3A conditions while CuE was not. We next analysed FAs (Fig. 7e–h) in cells with or without Sema3A and found that $Mn^{2+}$ rescued FAs number (Fig. 7f) and size (Fig. 7g, h) under Sema3A conditions. However, CuE was not able to rescue the effect of Sema3A and the rescue with $Mn^{2+}$ was partially abolished by adding CuE (Fig. 7f–h).

Is the ability of $Mn^{2+}$ to rescue single cell spreading sufficient to improve dispersion in explants? We plated explants on Fibronectin with or without Sema3A and let them migrate for 3 h (Fig. 7i). We found that adding $Mn^{2+}$ was sufficient to increase dispersion, as seen by the increased distance between the nearest neighbours (Fig. 7j). Then, we repeated the same assay but used increasing concentration of Sema3A, together with $Mn^{2+}$ or CuE (Fig. 7k), and checked the ability of explants to adhere (not washed away during fixation) and disperse (generating single cells). Explants showed a dose-dependent response to Sema3A in terms of adhesion and dispersion. Adding CuE lowered adhesion and dispersion in control explants (Fig. 7k) indicating that actin turnover is required for normal adhesion and dispersion in *Xenopus* NC cells. Adding $Mn^{2+}$ increased the rate of dispersing explants (Fig. 7k). Adding $Mn^{2+}$ to explants in the presence of Sema3A significantly increased the proportion of both adhering and dispersing explants (Fig. 7k). We then monitored the explants overtime to look at the dynamics of dispersion (Fig. 7l–m, Supplementary Movie 12). Explants exposed to Sema3A dispersed less than controls, adding $Mn^{2+}$ to the medium was sufficient to rescue dispersion. However, adding CuE lowered the effect of $Mn^{2+}$. Altogether, our data indicate that promoting cell-matrix adhesion via $Mn^{2+}$ is sufficient to rescue spreading in single cells and dispersion in explants and that normal actin dynamics is required for Mn-triggered rescue.

**Sdf1 requires Fibronectin for rescue in *Xenopus* NC cells.** *Xenopus* NC cells can adhere to Fibronectin, Laminin, Collagen and Vitronectin but only migrate on Fibronectin[37]. Interestingly, interaction between Sdf1 and Fibronectin is stronger than with laminin and collagens and promotes directional migration in other cell types[39]. We wondered whether the Sdf1/Sema3A competition we described here was depending on the fact that cells are cultured on Fibronectin. To test this idea, we cultured *Xenopus* NC cells on Matrigel (laminins and collagens) and analysed cell size, aspect ratio and circularity (Fig. 8a–d). *Xenopus* NC cells on Matrigel were less able to spread than cells on

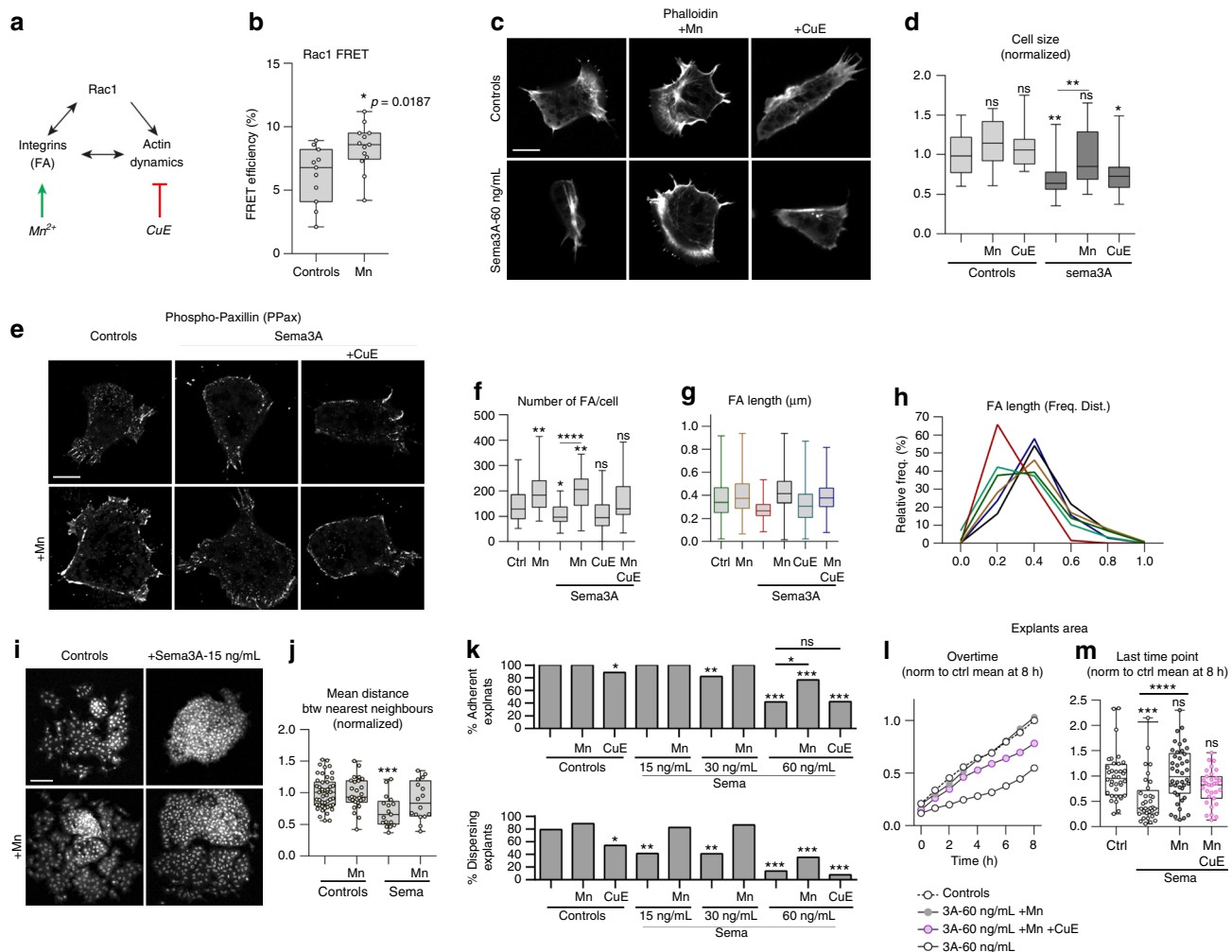

**Fig. 7** Activation of adhesion and actin dynamics are required for rescue of exposure to Sema3A. **a** Effects of Manganese ($Mn^{2+}$) and CucurbitacinE (CuE) on adhesion and actin. **b** Rac1 activity (FRET) in control/treated cells with $Mn^{2+}$ (2 mM) for 2 h, $n = 24$ cells ($n_{controls} = 11$, $n_{Mn2+} = 13$) from two independent experiments, unpaired Student $t$-test, $p$ value on the graph. **c** Cells cultured on Fibronectin with/without $Mn^{2+}$ (2 mM) or CuE (1 nM), Phalloidin. **d** Normalised cell area, $n = 104$ cells ($n_{controls} = 15$, $n_{Mn2+} = 17$, $n_{CuE} = 14$, $n_{Sema} = 20$, $n_{Sema+Mn2+} = 20$, $n_{Sema+CuE} = 18$), ANOVA, multiple comparisons; **$p_{(ctl\ vs\ sema)} = 0.0036$; **$p_{(sema\ vs\ sema+Mn)} = 0.0053$; and *$p_{(ctl\ vs\ sema+CuE)} = 0.0210$. **e** PPax immunostaining, cells cultured on Fibronectin with/without Sema3A at 60 ng/mL with/without $Mn^{2+}$ (2 mM) and/or CuE (1 nM). **f** Focal adhesion per cell per condition depicted in **e**, $n = 178$ cells ($n_{controls} = 29$, $n_{Mn2+} = 29$, $n_{Sema} = 30$, $n_{Sema+Mn2+} = 30$, $n_{Sema+CuE} = 30$, and $n_{Sema+Mn2++CuE} = 30$) from two independent experiments; ANOVA, multiple comparisons; *$p_{(ctl\ vs\ sema)} = 0.0348$; **$p_{(ctl\ vs\ Mn)} = 0.0053$; **$p_{(ctl\ vs\ sema+Mn)} = 0.0026$; ****$p_{(sema\ vs\ sema+Mn)} < 0.0001$. **g**, **h** FA length and frequency distribution from **e**, $n$ of individual FA analysed: $n_{controls} = 2197$, $n_{Mn2+} = 1688$, $n_{Sema} = 3101$, $n_{Sema+Mn2+} = 2708$, $n_{Sema+CuE} = 2469$, $n_{Sema+Mn2++CuE} = 2328$, from two independent experiments. Colour code same as in **g**. **i** Explants cultured on Fibronectin or Fibronectin plus Sema3A at 15 ng/mL with/without $Mn^{2+}$ (2 mM), stained with DAPI. **j** Mean distance between nearest neighbours from **i**, $n = 106$ explants ($n_{controls} = 47$, $n_{Mn2+} = 26$, $n_{Sema} = 17$, and $n_{Sema+Mn2+} = 16$), from two independent experiments; ANOVA, multiple comparisons, ***$p = 0.0005$. **k** Percentage of adherent and dispersing explants after a 3-hour culture on Fibronectin or Fibronectin plus Sema3A at 15, 30 or 60 ng/mL with/without $Mn^{2+}$ (2 mM) and/or CuE (1 nM), $n = 268$ explants ($n_{controls} = 52$, $n_{Mn2+} = 26$, $n_{CuE} = 26$, $n_{Sema15} = 17$, $n_{Sema15+Mn2+} = 17$, $n_{Sema30} = 22$, $n_{Sema30+Mn2+} = 22$, $n_{Sema60} = 43$, $n_{Sema60+Mn2+} = 17$, and $n_{Sema60+Mn2++CuE} = 26$) from five independent experiments. Comparisons of proportions were made using contingency tables[60]. Null hypothesis rejected if T > 3.841 (*$\alpha = 5\%$); T > 6.635 (**$\alpha = 1\%$); T > 10.83 (***$\alpha = 0.1\%$). **l** Normalised area overtime for explants cultured on Fibronectin or Fibronectin plus Sema3A at 60 ng/mL with/without $Mn^{2+}$ (2 mM), $n = 145$ explants ($n_{controls} = 38$, $n_{Sema} = 33$, $n_{Sema+Mn} = 42$, and $n_{Sema+Mn+CuE} = 32$) from four independent experiments. **m** Normalised explant areas after 8 h for conditions shown in **l**; ANOVA, multiple comparisons, ***$p = 0.0005$; ****$p < 0.0001$. Scale bars (**c**, **e**), 10 μ. Scale bar in **i**, 100 μ. Box and whiskers plot: the box extends from the 25th to the 75th percentile; the whiskers show the extent of the whole dataset. The median is plotted as a line inside the box. Source data are provided as a Source Data file

Fibronectin (Fig. 8a–b). Sdf1 alone had no effect (Fig. 8a–d). However, Sema3A strongly inhibited adhesion to Matrigel. Most cells were lost during fixation and the few remaining cells were round (Fig. 8c, d), with no obvious actin filaments (Fig. 8a). Adding Sdf1 to cells exposed to Sema3A on Matrigel slightly improved spreading (Fig. 8b), aspect ratio (Fig. 8c) and circularity (Fig. 8d) in the very few cells that remained attached. None of these parameters were rescued to control levels. Adding Sdf1 only

partially prevented detachment from the Matrigel as most cells were still lost during fixation. These results indicate that Sema3A affects adhesion and spreading on Matrigel as it does on Fibronectin but that Sdf1 is only able to counterbalance Sema3A on Fibronectin (see Fig. 4c–e).

To check whether Sdf1's activity systematically depends on Fibronectin, we decided to use the mouse cephalic NC cell line, O9-1[40]. We confirmed by PCR that O9-1 cells expressed *Nrp1*,

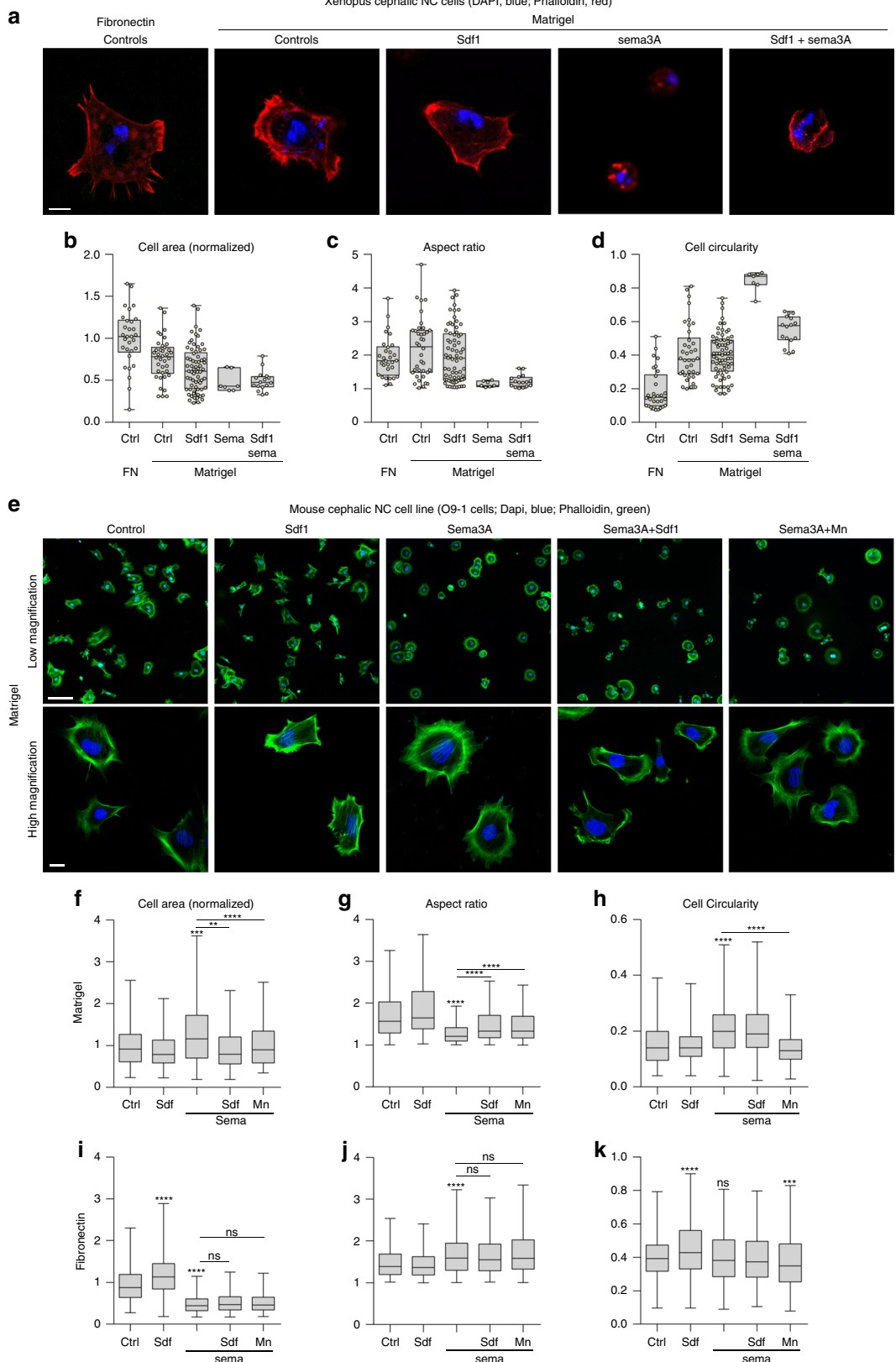

*Nrp2* and *Cxcr4* together with NC markers *Sox9* and *Ets1* (Supplementary Fig. 8). We cultured O9–1 cells on Matrigel with or without Sema3A and/or Sdf1 or $Mn^{2+}$ and analysed cell size, aspect ratio and circularity (Fig. 8e–h). Sdf1 alone had no significant effect on these cell parameters. However, Sema3A slightly increased spreading (Fig. 8f) and circularity (Fig. 8h) and

reduced the aspect ratio (Fig. 8g) indicating that cells were less polarised. The actin cytoskeleton was organised as a large circle surrounding the nucleus (Fig. 8e) instead of being accumulated on one side of the cell with filaments in protrusions, as seen in control conditions. Interestingly, adding Sdf1 or $Mn^{2+}$ was sufficient to rescue spreading (Fig. 8f), aspect ratio (Fig. 8g) and

**Fig. 8** Sdf1 does not need to interact with Fibronectin to rescue the effect of Sema3A. **a** *Xenopus* NC cells cultured on Fibronectin or Matrigel with or without Sema3A coated at 60 ng/mL and/or Sdf1 added in solution at 0.5 µg/mL, stained with DAPI and Phalloidin. **b–d** Normalised cell area (**b**), aspect ratio (**c**) and circularity (**d**) per cell for each condition shown in (**a**). For **b**, **c** and **d**, $n = 160$ cells (FN: $n = 30$, Matrigel, $n_{controls} = 38$, $n_{Sdf} = 69$, $n_{Sema} = 7$ and $n_{Sdf+Sema} = 16$). Since only seven cells remained attached in the Sema3A condition we did not perform statistical analysis. **e** Mouse neural crest cell line, O9-1, cultured on Matrigel with or without Sema3A coated at 60 ng/mL and/or Sdf1 at 0.5 µg/mL and/or $Mn^{2+}$ (2 mM) added in solution, stained with DAPI and Phalloidin. **f–h** Normalised cell area (**f**), aspect ratio (**g**) and circularity (**h**) per cell for each condition shown in **e**. $n = 1334$ cells ($n_{controls} = 370$, $n_{Sema} = 268$, $n_{Sdf} = 96$, $n_{Sema+Mn} = 224$ and $n_{Sema+Sdf} = 376$) from four independent experiments. ANOVA followed by multiple comparisons, ****$p < 0.0001$; ***$p < 0.001$; **$p < 0.01$. **i–k** Normalised cell area (**i**), aspect ratio (**j**) and circularity (**k**) per cell for mouse NC cells cultured on Fibronectin with or without Sema3A coated at 60 ng/mL and/or Sdf1 (0.5 µg/mL and/or $Mn^{2+}$ (2 mM) added in solution corresponding to experimental conditions shown in Supplementary Fig. 8. $n = 3421$ cells ($n_{controls} = 653$, $n_{Sema} = 804$, $n_{Sdf} = 675$, $n_{Sema+Mn} = 591$ and $n_{Sema+Sdf} = 698$). ANOVA followed by multiple comparisons, ****$p < 0.0001$; ***$p < 0.001$. Scale bars in **a**, 10 µ; in **e** low magnification, 50 µ; in **e** high magnification, 10 µ. Box and whiskers plot: the box extends from the 25th to the 75th percentile; the whiskers show the extent of the whole dataset. The median is plotted as a line inside the box. Source data are provided as a Source Data file

circularity (Fig. 8h) and to restore local distribution of actin filaments associated with protrusions (Fig. 8e, high magnification). The fact that Sdf1 rescues the effect of Sema3A in mouse NC cells on Matrigel indicates that Sdf1-Fibronectin interaction is not a pre-requisite for Sdf1's function *per se*. Next, we tested the Sdf1/Sema3A competition in mouse NC cells on Fibronectin (Fig. 8i–k, Supplementary Fig. 8). Sdf1 and Sema3A had opposite effects on spreading (Fig. 8i). However, Sema3A had only little effect on the aspect ratio and circularity (Fig. 8j, k). Neither Sdf1 nor $Mn^{2+}$ were able to rescue spreading of mouse NC cells in the presence of Sema3A and Fibronectin. Altogether these data indicate that while Sdf1 is technically able to rescue exposure to Sema3A irrespectively of the substrate type, in *Xenopus* NC cells, Fibronectin is required.

**Global activation of adhesion rescues lack of Cxcr4 in vivo**. Next, we assessed Fibronectin distribution in vivo. It was previously reported that, at trunk level, Fibronectin is lacking above the dorsal midline and only later assembled above the neural tube[41] but no equivalent study at cephalic levels has been performed. At stage 17, Fibronectin is found underlying the neural plate, the NC domain and the ectoderm, around the notochord and beneath the lateral mesoderm (Fig. 9a). Interestingly, no Fibronectin is observed in between the neural plate, the NC and the superficial pigmented layer (Fig. 9a). At stage 20, when migration has just started, Fibronectin is still absent dorsally to the neural plate but is now seen between the NC and the ectoderm. In addition, Fibronectin starts being deposited at the interface between the neural plate and NC cells (Fig. 9b). These results indicate that at the onset of *Xenopus* cephalic NC migration, Fibronectin is pre-dominantly located ventro-laterally. Could such bias of Fibronectin distribution, together with a strong dorsal expression of *Sema3A* (Fig. 1a–c) and the presence of epithelial structures at the midline such as the neural plate and the notochord, be enough to drive directional migration in absence of Sdf1 signalling?

To test this hypothesis, we performed grafting experiments with control cells or Cxcr4-MO cells. NC cells were grafted directly into control embryos or pre-incubated with $Mn^{2+}$ for 30 min before grafting (Fig. 9c). Control cells grafted into control embryos migrated normally (Fig. 9d, e, arrowheads) while Cxcr4-MO cells did not (Fig. 9d, e, asterisk). Surprisingly, exposure to $Mn^{2+}$ prior to grafting significantly restored directional migration of Cxcr4-MO cells in vivo (Fig. 9d, e, arrows). This is a striking result. It demonstrates that the local environment is sufficient to bias NC migration towards ventral regions in absence of Cxcr4/Sdf1 signalling. To substantiate this result, we attempted to rescue Cxcr4MO by activating Rac1 via Tiam1 photoactivation (Fig. 9f–h). NC cells injected with Cxcr4MO and CIBN alone or CIBN together with Tiam1 were grafted into control embryos.

Embryos were illuminated with a LED at 470 nm every 10 min to generate a transient but recurrent Rac1 activation. Importantly, the whole embryo was illuminated to avoid generating any bias in Rac1 activation. Cxcr4MO/CIBN cells mostly stayed near the region of the graft (Fig. 9f, Supplementary Movie 13) whereas cells with Cxcr4MO, CIBN and Tiam1 migrated towards ventral regions of the face (Fig. 9g, h). These data indicate that promoting a global Rac1 activity in NC cells unable to respond to Sdf1 is sufficient to promote dorso-ventral migration.

**Discussion**

The results of our study indicate that premigratory NC cells are surrounded by Class3-semaphorins, with a strong dorsal expression of Sema3A, and that Sdf1 and Fibronectin are predominantly present in ventro-lateral regions (Fig. 10a). Class3-Semaphorins reduce adhesion to Fibronectin and dispersion. On the contrary, cells exposed to both Sdf1 and Semaphorins can efficiently spread and migrate. These data, together with the ability to rescue directional migration of Cxcr4-MO cells with global $Mn^{2+}$ treatment or unbiased Rac1 activation, led us to propose that the initiation of directional migration is primarily linked to the uneven organisation of the cephalic region at this stage of development. There is a biased distribution of Fibronectin, a strong dorsal expression of Sema3A and physical obstacles such as the neural plate and the notochord located at the midline (Fig. 10b). The competition for the control of cell-matrix adhesion at the single cell level (between Sema3A and Sdf1) can be translated into directional migration due to the non-homogenous organisation of the complex 3D environment. Importantly, it suggests that oriented topology may render precise gradients of positive or negative cues dispensable.

Expression of *Class3-semaphorins*, *Sdf1* and *Cxcr4* precedes the onset of NC migration by several hours. The onset of NC migration is linked to an increase of the stiffness of the mesoderm located underneath the NC domain[42]. Interestingly, NC migration can start earlier if the local stiffness is experimentally increased. Matrix rigidity is known to feedback into FAs[43]. Thus, one could test whether an experimental increase of matrix stiffness might be sufficient to compensate for a lack of Sdf1 in vivo.

It would be interesting to explore whether this topology-biased mechanism is also present during *Xenopus* gastrulation. Sdf1 is important for *Xenopus* gastrulation and a gradient of Sdf1 can attract mesodermal cells in vitro[44,45]. However, *Sdf1* is broadly expressed in the ectoderm overlying the gastrulating mesoderm[44] while Fibronectin distribution is not homogenous[41]. Fibre density is higher in the middle part of the blastocoel roof than under the early migrating mesoderm[41] raising the possibility that mesodermal cells may move from low to high Fibronectin concentrations instead of following an Sdf1 gradient.

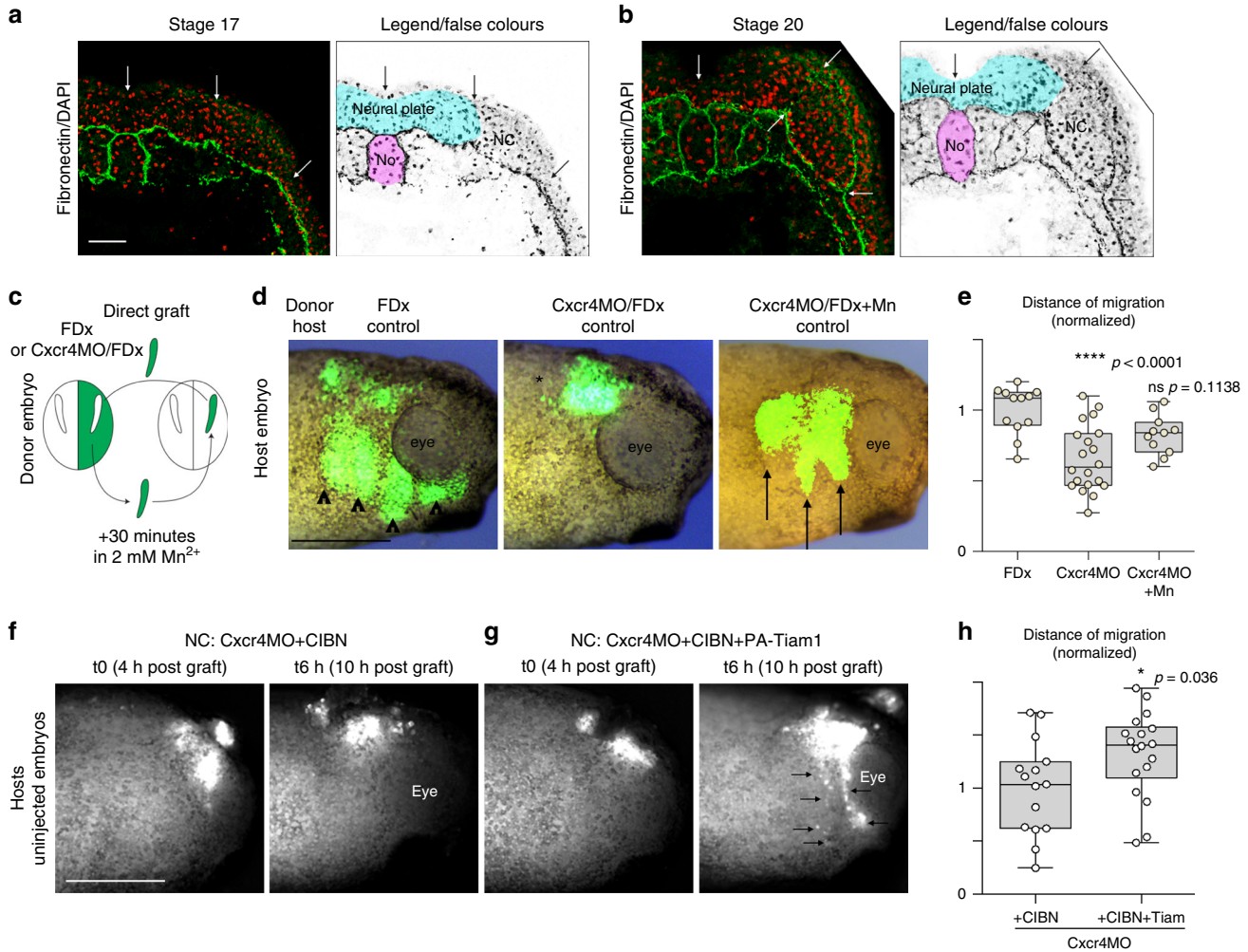

**Fig. 9** Global activation of adhesion in vivo is sufficient to rescue Cxcr4 loss-of-function. **a**, **b** Fibronectin immunostaining on cryosections through the cephalic region of St17 (**a**) and St20 (**b**) *Xenopus laevis* embryos. **c** Diagram depicting the grafting procedure. **d** Representative images of the three types of grafts that were performed. Controls NC cells (FDx), cells injected with Cxcr4MO with or without prior exposure to $Mn^{2+}$ were grafted into control non-injected hosts embryos. **e** Normalised net distance of migration along the dorso-ventral axis of grafted cells after an overnight incubation following the graft, $n = 43$ grafted embryos from four independent experiments. ANOVA followed by multiple comparisons, $p$ values are indicated on the graph. **f** Still images from time-lapse movies for Cxcr4MO cells co-injected with CIBN-GFP grafted into control embryos. **g** Still images from time-lapse movies for Cxcr4MO cells co-injected with CIBN-GFP and photoactivatable Tiam1 grafted into control embryos. Arrows indicate NC cells migrating along the dorso-ventral axis. **h** Normalised net distance of migration along the dorso-ventral axis of grafted cells after 6 h of time-lapse imaging, $n = 33$ grafted embryos from five independent experiments. Student's *t*-test, two tailed, $p$ value indicated on the graph. FDx, fluorescein dextran; NC, neural crest; No, notochord. Scale bar in **a**, 50 μ. Scale bars in **d** and **f**, 500 μ. Box and whiskers plot: the box extends from the 25th to the 75th percentile; the whiskers show the extent of the whole dataset. The median is plotted as a line inside the box. Source data are provided as a Source Data file

VEGFA is essential for chick NC migration and an ectopic source of VEGFA is sufficient to deviate NC migration towards Semaphorin-rich domains[21,22]. Thus, VEGFA was proposed to act as a gradient despite its homogenous distribution along the lateral ectoderm. VEGFA loss-of-function does not prevent early migration but blocks NC cells at the entrance of the branchial arches, a structure expressing *Sema3F*. Interestingly, VEGFA binds more to Fibronectin at acidic than neutral pH[46]. Since cephalic NC migration in chick occurs in hypoxia[47] the pH is likely to be acidic due to anaerobic metabolism. Vascularisation arrives in the branchial arches at stage HH12 which corresponds to the entry of NC cells into the arches. The arrival of blood supply likely brings back normoxia and neutral pH values and may favour the release of VEGFA. Therefore, an hypothesis is that entry of NC cells into the arches is controlled by vascularisation/pH-dependent release of VEGFA. VEGFA could compete with Semaphorins since Nrp1 is a co-receptor for VEGFA and

Sema3A or by antagonistic effects on downstream effectors. Interestingly, a competition between Semaphorin and VEGF signalling has been shown in corneal development[48].

Here we show that Sema3A inhibits Rac1 and RhoA. However, in growth cones Sema3A induces Rac1[49] and activates the Rho/ROCK/Myosin II pathway[50–53]. In addition, dynamics of small GTPases upon Sema3A are extremely complex in growth cones exemplified by the dynamics of waves of Cdc42 and RhoA[54]. Further, adhesion of growth cone to the substrate and the turn-over of focal contacts are extremely sensitive to variation of Rac1 and RhoA levels[50]. Activation or inhibition of either of these GTPases have negative effects on adhesion. Furthermore, data on growth cones come from various species and cell types (dorsal root ganglia from mouse or chicken embryos, *Xenopus* retinal ganglia, rat PC12 cell line induced to form neurites) cultured on Laminin (+/−poly-lysine) while *Xenopus* NC are cultured on Fibronectin. Since Rac1 levels depend on the multiple inputs that

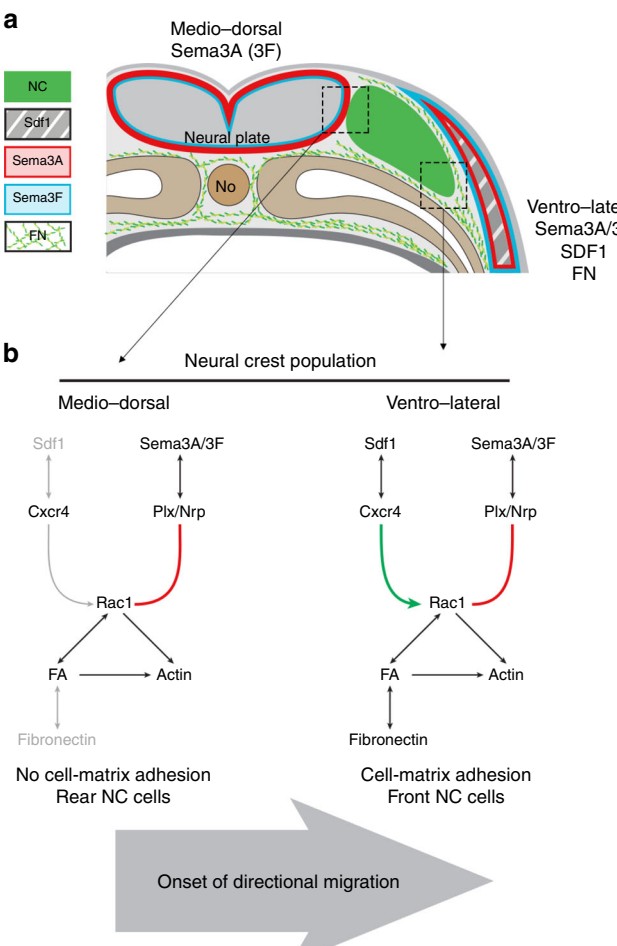

**Fig. 10** Model. **a** Diagram summarising the expression of *Sema3A* (red), *Sema3F* (blue), *Sdf1* (grey, stripped), and the distribution of Fibronectin (green fibres) on a transversal section at the onset of NC cell migration (NC cells are in green). **b** Diagram of the proposed signalling events taking place. Sdf1 activates Rac1 whereas Sema3A inhibits Rac1. Medio-dorsally, Fibronectin and Sdf1 are lacking. Thus, the effect of Sema3A on Rac1 dominate and NC cells cannot adhere to the matrix. Ventro-medially, all players are present. Sdf1 counterbalances Sema3A, NC cells can adhere to Fibronectin. No, notochord; Plx/Nrp, plexins/neuropilins

control it (growth factor, guidance cues, adhesion–related signalling) it is difficult to directly compare the effect of one molecule (here Sema3A) in different cell types cultured on different substrates.

In Sema3A-KD embryos, NC streams are shorter and numerous NC cells are located in ectopic places. Yet, migration occurs along the dorso-ventral path and streams still form. This indicates that Sema3A and 3F are only partially involved in the patterning of NC migration in *Xenopus*. One possibility to explain this result may be via the known interaction between NC cells and epibranchial placodes. NC and placodes undergo a coordinated collective migration based on Sdf1 chemotaxis and contact-inhibition of locomotion, coined chase-and-run[20] leading to the accumulation of epibranchial aggregates in between NC streams. These placodal structures act as physical barriers preventing NC cells crossing from one stream to another. Therefore, NC streams still emerge regardless of the absence of Sema3F, which is expressed in epibranchial placodes. By contrast, in chicken, where a loose mesenchyme is present underneath the placodal regions, an inhibition of Class3-semaphorins leads to an immediate invasion of NC cells in between the streams[55,56].

Sdf1, Sema3A and Sema3F are involved in melanoma, multiple myeloma, glioblastoma, neuroblastoma, pancreatic, prostate, ovarian and lung cancers[57–59]. Since these pathways are being proposed as putative therapeutic targets, it is important to consider that they may antagonise each other and that their functions may not be systematically related to a local organisation as a gradient or even linked to attraction and repulsion of migratory cells.

## Methods

**Reagents and solutions.** Agarose (Fischer Scientific, BP1356–100), Bovine Serum Albumin (Sigma, A4503), Fibronectin (Sigma, F1141), dextran fluorescein (Invitrogen, D1820), dextran tetramethylrhodamine (Molecular Probes life technologies, D3312), Gelatin (Sigma, G1890), Methylcellulose (Sigma, M0387–100), Mowiol 40–88 (Fluka, 81386), Penicillin-Streptomycin (Sigma, P4458), Sucrose (VWR, 27480.294). Danilchick's medium 1 × (DFA): NaCl (53 mM), NA2CO3 (5 mM), K Gluconate (4.5 mM), Na Gluconate (32 mM), MgSO4–7H20 (1 mM), CaCl2 (1 mM), BSA 0.1%. MEMFA: MOPS (1 mM), EGTA (2 mM), MgSO4 (1 mM), Formaldehyde (3.7%). Normal Amphibian Medium (NAM): NaCl (110 mM), KCl, (2 mM), Ca(CO3)2 (1 mM), MgSO4 (1 mM), EDTA (0.1 mM), NaHCO3 (1 mM), Sodium Phosphate (2 mM).NTMT: NaCl (0.1 M), TrisHCl pH9.5 (0.1 M), MgCl2 (50 mM), Tween 0.1%. Phosphate Buffer Saline 1 ×: NaCl (137 mM), KCl (2.7 mM), Na2HPO4 (10 mM), KH2PO4 (1.8 mM), CaCl2–2H2O (1 mM), MgCl2–6H2O (0.5 mM). Human Sdf1 (Calbiochem, 572300; used at 0.5 µg/mL in solution). Mouse Semaphorin-3A-Fc (R&D Systems, 5926-S3; used at 15, 30 or 60 ng/mL coated or added in solution), mouse Semaphorin-3F-Fc (R&D Systems, 3237-S3; used at 120, 240 or 480 ng/mL coated).

**Ethics statement.** Male and female *Xenopus laevis* were used to produce fertilised eggs. Animal care and experimentation were conducted in accordance with institutional and national guidelines, under the institutional licence number A 31 555 01 delivered by the Préfecture de la Haute-Garonne running until March 2019.

**Microinjections of *Xenopus laevis* embryos.** All transfections were performed by microinjections with pulled glass needles at 8-cell stage.

**Statistical analysis.** Comparison of percentages was performed using contingency tables[60]. Two data sets were considered significantly different (null hypothesis rejected) if T > 3.841 (*α = 0.05), T > 6.635 (**α = 0.01) or T > 10.83 (***α = 0.001). Normality of data sets was tested using Kolmogorov-Smirnov's test, d'Agostino and Pearson's test and Shapiro-Wilk's test using Prism6 (GraphPad). A data set was considered normal if found as normal by all three tests. Datasets following a normal distribution were compared with Student's *t*-test (two-tailed, unequal variances) or a one-way ANOVA with a Dunnett's multiple comparisons post-test in Prism6 (GraphPad). Datasets that did not follow a normal distribution were compared using Mann Whitney's test or a non-parametric ANOVA (Kruskal–Wallis with Dunn's multiple comparisons post-test) using Prism6 (GraphPad). Cross-comparisons were performed only if overall P value of the ANOVA was <0.05. Strategy for sample size determination do not apply here since all embryos or cells available were analysed. Statistics were performed on the whole population. No data were excluded from analysis apart for very large data sets (>1000 cells) on 09–1 cells for which automatic outlier detection was used. Variances were not assumed to be equal. Box and whiskers plot: the box extends from the 25th to the 75th percentile; the whiskers show the extent of the whole dataset. The median is plotted as a line inside the box. Statistics are provided in figure legends or added directly on the graphs. All error bars on graphs and curves that are not box and whiskers plots correspond to the standard deviation (s.d) or standard error of the mean (s.e.m) as indicated in figure legends.

**Cell tracking.** Cell tracking on bright field movies was performed with the Manual Tracking plug-in in ImageJ. Spot tracking function in Imaris was used to follow individual cells transfected with nuclear-mCherry. EB3 comets were tracked using spot tracking function in Imaris. Directionality, velocity, track length or duration were extracted using the Chemotaxis Tool plug-in in ImageJ or calculated from data generated by Imaris tracking tool.

**Neural crest cultures.** Neural crest cells were isolated from stage 18 embryos using an eyebrow knife (eyebrow mounted on a glass pipette) and plated on Fibronectin-coated Ibidi µ-slides dishes (ref 80821). Dishes were prepared by incubating Fibronectin solution at 10 µg/mL for one hour at 37°. When coating with semaphorins, semaphorins are coated first for 1 h at 37 °C, then Fibronectin is coated on top.

**Analysis of dispersion and cell shape descriptors.** Dispersion of neural crest explants was measured by drawing a line between all cells located at the periphery of an explant. The total area occupied by the cells and the free space in between are included so as to represent to total spreading of the explant. Cell shape descriptors

(area, circularity, aspect ratio) were obtained by drawing cell's contour in FIJI. Images of membrane tracer or Phalloidin staining were thresholded and contours were detected using the magic wand. Values were retrieved using cell shape descriptor's tool in FIJI. For PPax area images were converted to 32-bit before thresholding. Empty pixels were converted as NaN to be excluded from the total area calculation. Area occupied by PPax is divided by total area occupied by the cell.

**Neural crest grafts**. The appropriate Morpholino oligo and fluorescein dextran (FDx) or rhodamine dextran (RDx) were injected into 8-cell stage embryos. Embryos at stage 18 are immobilised into a Petri dish filled with plasticine. The pigmented ectoderm layer located above the NC region is carefully removed. NC cells are mechanically detached from their surrounding tissues by applying gentle pressure on the side of the neural crest domain in a lateral to medial direction. To perform the graft a given NC explant has first to be taken out of a host embryo, then a NC explant is taken from the donor embryo and placed in the wound of the host and kept in place by a piece of glass coverslip for 15 min. The coverslip is then removed and embryos are left to heal. Further details on grafting procedures can be found elsewhere[20,61].

**In situ hybridisation**. Embryos were fixed overnight at 4 °C in MEMFA before being dehydrated by several baths in pure methanol. Embryos are then rehydrated by solutions of decreasing methanol concentration, washed in PBS and bleached in a peroxide solution to remove skin pigments. After bleaching, a short post-fixation in formaldehyde 3,7% is performed. Embryos are then processed using the InsituPro VS (Intavis AG Bioanalytical Instruments, Germany) automate. Briefly, embryos are incubated 16 h at 65 °C in formamide-based hybridisation buffer containing a digoxigenin-labelled antisense probe against the gene of interest. Probes are washed in formamide-based washing solutions, then washed in PBS 0.1% tween, incubated in a serum-based blocking solution for 1 hour and incubated for 1 hour in blocking solution containing the anti-digoxigenine antibody coupled with alkaline phosphatase (Roche, 11093274910, 1/2000). Staining is achieved by incubating embryos in a pH9.5 solution containing NBT (Promega, S380C) at 50 μg/mL and BCIP (Promega, S381C) at 100 μg/mL. The following probes were used: XL-Sdf1[62], XL-Cxcr4[63], XL-Sema3A[24], XL-Sema3F[24], XL-snail2[64] and XL-Twist1[65].

**Immunostaining on cell cultures**. *Xenopus* NC cells were cultured on Fibronectin-coated dishes, left to migrate for a few hours, then fixed in PFA 4% for 30 min, blocked and permeabilized in PBS1X/2%serum/0.1%Triton for 30 min and incubated 2 h at room temperature or overnight at 4 °C with a primary antibody, washed in PBS and incubated 1 hour at room temperature or overnight at 4 °C with a secondary antibody mixed with DAPI or Phalloidin if necessary. Primary antibodies: rabbit anti-Phospho-Paxillin Tyr118 (Upstate, 07–733; 1/200), mouse anti-Rac1-GTP (NewEast, 26903; 1/500). Secondary antibodies: goat anti-rabbit Alexa-488 or 555, goat anti-mouse IgM-594 (Invitrogen; all used 1/1000). Counter-staining was done with DAPI and Phalloidin coupled with Alexa-488, 555 or 633 (all diluted 1/1000).

**Cell culture, drugs and proteins**. Mouse neural crest cell line O9–1[40] was purchased from Merck Millipore (ref SCC049) and cultured in DMEM supplemented with 15% foetal bovine serum (FBS, Sigma F7524), 0.1 mM minimum essential medium (MEM) non-essential amino acids (Gibco™ 11140050), 100 U/mL penicillin, 100 μg/mL streptomycin. *Xenopus* cranial NC cells were dissected as described in (DeSimone et al., 2005). O9–1 cells were not tested for contamination by mycoplasma. *Xenopus* NC cells primary culture. Briefly, at stage 18 the pigmented epidermal layer is removed then NC cells are gently taken out by micro-dissection. *Xenopus* NC cells are cultured in Danilchick's modified medium (DFA). When needed, cell dissociation was performed by putting the NC explants in $Ca^{2+}/Mg^{2+}$-free medium for 10 min before transferring them to DFA. Fibronectin (Sigma, F1141) coating was done by incubating the dish with a 10 μg/mL solution for one hour at 37 °C for plastic dishes, at 100 μg/mL one hour at 37 °C for glass-bottom dishes. Matrigel coating was done by incubating Matrigel (Corning® 354234) at 0.18 mg/mL for one hour at Room Temperature. Purified mouse Sema3A/Fc (R&D Systems, 5926–53) and Sema3F/Fc (R&D Systems, 3237–53) coating were done by incubating 15–60 ng/mL (3 A) 120–480 ng/mL (3 F) solution for one hour at 37 °C. See main text and figure legends for details. Sdf1 (Calbiochem, 572300), $Mn^{2+}$ (from Sigma M7634, MnSO4, 1 M stock solution), Cucurbitacin E (Sigma, SML0577), and DMSO (Sigma, D8418) were used in solution. See concentrations in the main text and the figure legends. The following dishes were used: Ibidi μ-slide 8-well plates (80821), Lab-Tek 4-chamber #1.0 borosilicate coverglass system (155383).

**Photoactivatable Tiam1**. CIBN-caax-GFP and PA-Tiam1-CRY2-mCherry were coinjected with a stoichiometry of 5:1. Photoactivation is performed with any light below 500 nm. We used a 470 nm LED on an epifluorescence Nikon AxioImager equipped with a Colibri light source or a 488 nm laser on an inverted Leica SP8 confocal or an inverted Zeiss 710 confocal. Pulses of hundreds of milliseconds are

enough to drive membrane localisation of PA-Tiam1. There is a 2–3 min gap between photoillumination and the formation a new lamellipodia at the site of photoillumination.

**CRISPR/Cas9 Loss-of-function**. CRISPR-Cas9-mediated gene editing was adapted from Vejnar et al.[66], specific details are given below.

**Target sites selection**. Briefly, target sites were selected using the CRISPRscan (crisprscan.org) tool, with major criteria that sgRNAs score the highest and above 55, have no predicted off-targets, and preferentially target both the S and L forms of each target sequence (Xenbase.org; X.*laevis* 9.2). Three target sites were chosen for Sema3A and Sema3F.

**sgRNA generation**. First, sgRNA DNA templates were synthesised by fill-in PCR with the universal primer [same ref] and each of the six specific primers (5′− 3′) obtained on CRISPRscan:

Sema3a _sg1 taatacgactcactataGGTGATGATTTCCAAGCGTGgttttagagctagaa;
Sema3a _sg2 taatacgactcactataGGATGTTGGCCACTCAGAGAGCCCAT-3′;
Sema3a _sg3 taatacgactcactataGGGGGCTGGAAAAGATATTGgttttagagctagaa;
Sema3f _sg1 taatacgactcactataGGGCGGTGCACTGGATACACgttttagagctagaa;
Sema3f _sg2 taatacgactcactataGGTTGTATGCGTGGAGTGGGgttttagagctagaa;
Sema3f _sg3 taatacgactcactataGGCTGTACGCGTGGAGTGGGgttttagagctagaa.

Second, the synthesised DNA templates were used to produce sgRNAs by in vitro transcription reaction (MaxiScript T7 kit, Ambion, AM1314).

**CRISPR/Cas9 knockdown**. A cocktail of three sgRNAs (500 pg each) targeting Sema3A or Sema3F were co-injected together with 36 ng of either EnGen® Cas9-NLS (NEB #MO646M) or Cas9-Dead-NLS (Genaxxon Bioscience) proteins, 0.1 M KCl and fluorescein dye.

All embryos were then analysed by in situ hybridisation or used for grafting experiments. We did not keep some embryos to assess the occurrence of genomic modifications.

**Expression vectors, Morpholinos**. Life-Act-GFP[67], EB3-GFP[68], CIBN-Caax-GFP[34], CRY2-Tiam1-mCherry[34]. Rac1, RhoA and Cdc42 FRET probes[69].

Following antisense Morpholinos were purchased from Gene-Tools
Standard control oligo: 5′-CCTCTTACCTCAGTTACAATTTATA-3′
Anti-Sema3A MO: 5′-ATGCAATCCAGGTCAGAGAGCCCAT-3′
Anti-Sema3F MO: 5′ -GGAACGCAGAAGGACACCCA − 3′
Anti-Cxcr4 MO[19]: 5′-CAATGCCACCAGAAAACCCGTCCAT-3′

**Histology**. For cryosections, embryos were incubated overnight in Phosphate Buffer (PB)/15% sucrose at 4 °C. Then embryos were passed in PB/15% sucrose/7.5% gelatine for 2 h at 42 °C before being embedded in PB/15% sucrose/7.5% gelatine. Cryostat Leica CM1950 was used to generate 20-μ-thick sections.

**FRET analysis**. The CFP and YFP channels were excited using the 440-nm diode laser and the 514-nm argon lines, respectively. The two emission channels were split using a 545-nm dichroic mirror, which was followed by a 475–525-nm bandpass filter for CFP and a 530-nm longpass filter for YFP (Chroma Technology Corp). Pinholes were opened to give a depth of focus of 2 mm for each channel. Scanning was performed on a sequential line-by-line basis for each channel. The gain for each channel was set to 75% of dynamic range (12-bit, 4096 grey levels) and offset such that backgrounds were zero. Time-lapse mode was used to collect one prebleach image for each channel followed by bleaching with a minimum of 50 iterations of the 514-nm argon laser line at maximum power (to bleach YFP). A second postbleach image was then collected for each channel. Control nonbleached areas were acquired in the same field of view as bleached cells to confirm specificity of FRET detection. Pre and post-bleach CFP and YFP images were then imported into ImageJ for processing (AccpbFRET plugin[70]; - %FRET = $(I_{donor-postbelaching} − I_{donor-prebleaching})/I_{donor-postbleaching}×100\%$). Within this, all pixels were reregistered (using the Fast Hartley Transform algorithm) to correct for any shifts in the xy plane; bleed-through between donor and acceptor channels and efficiency of acceptor photobleaching were all accounted for in the analysis.

**PCR on O9–1 cells**. O9 cells were collected on passage 3 and 5 after trypsin treatment and washed in cold PBS. Cell were centrifuged and used for RNA extraction using RNeasy Plus Micro Kit (Qiagen). cDNA was synthesised on 1 μg of total RNA using superscript II according to manufacturer instructions. PCRs were performed using GoTaq polymerase on 25 ng cDNA per reaction to detect Cxcr4, Cxcr7, Nrp1, Nrp2, Sox9 and Ets1. The original image of the gel used for sup Fig. 8 is provided in the source data file.

Following primers were used:
Neuropilin 1 (NM_008737.2), forward TGTAGGGACACAGGGTGCCA, reverse GCCTTGCGCTTGCTGTCATC
Neuropilin 2 (NM_001077403.1) Fwd CATGCTGGGGATGCTCTCGG, Rev TTTTCCCCACACTCGGTGGC

Cxcr4 (NM_009911.3) Fwd TCGCTATTGTCCACGCCACC, Rev GCTGGA GCCTCTGCTCATGG

Sox9 (NM_011448.4) Fwd GGCAGACCAGTACCCGCATC, Rev AGTGTAG GTGACCTGGCCGT

Ets1 (NM_011808.2) Fwd CACAGGAAGTGGGCCGATCC, Rev CTCAGGA GCTATTGCCCCGC

**Reporting Summary**. Further information on experimental design is available in the Nature Research Reporting Summary linked to this article.

## Data availability

All relevant data are available from the authors upon reasonable request.

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

## Acknowledgements

This work was supported by grants from Midi-Pyrénées Regional Council (Installation Grants for Excellent Researchers, 13053025), Fondation pour la Recherche Medicale (FRM, AJE201224), Toulouse Cancer Santé (DynaMeca) the CNRS and Université Paul Sabatier to E.T. and grants from the Medical Research Council (M010465 and J000655) and Biotechnology and Biological Sciences Research Council (M008517) to R.M.; F.B. is supported by a post-doc fellowship from the Midi-Pyrénées Regional Council (grant 13053025), N.G. is a recipient of an individual fellowship from FRM (ARF20150934153) and the Marie Curie Prestiges Program (PRESTIGES 2015–4–007). We are grateful to Oriol Viader for sharing data that contributed to Supplementary Fig. 1. We thank Britta Eickholt (Charité University, Berlin; Germany) for providing chick Sema3A conditioned medium that was used to performed preliminary experiments for this project. We thank Mathieu Coppey (Institut Curie, Paris; France) for providing CRY2/CIBN photo-activatable vectors and Nicolas David (Ecole Polytechnique, Palaiseau; France) for providing pCS2+/Life-Act-GFP.

## Author contributions

E.T. and R.M. conceived the project. F.B. and E.T. designed and performed most of the experiments. E.T. and M.P. performed and analysed the FRET experiments. N.G. and F.B. performed the O9 cells experiments. C.C. and E.T. performed the cell adhesion assay. F.B., E.T., C.C. and R.M. analysed and interpreted the data. E.T. and F.B. prepared the figures and supplementary materials. E.T. wrote the paper. All authors commented on the manuscript.

## Additional information

**Competing interests:** The authors declare no competing interests.

