## [Peer Review File · Nature Communications]

Reviewers' comments:

Reviewer #1 (Remarks to the Author):

Combining the analysis of cephalic neural crest cell (NCC) migration in Semaphorin 3A (Sema3A), Sema3F, and C-X-C chemokine receptor type 4 (Cxcr4; coupled or not with Mn²⁺) *Xenopus* morphants with the expression pattern of Stromal cell-derived factor 1 (Sdf1), Sema3A, Sema3F, and fibronectin, Bajanca and colleagues hypothesize a model in which the appearance of NCC streams would not depend on the existence of stripe-patterned repulsive Sema3s combined with Sdf1-expressing areas located at a distance. Largely based on in vitro data, the Authors propose instead that cephalic NCC streams would arise because of the asymmetric distribution of fibronectin combined with the presence of non-patterned Sdf1 and Sema3s that would, respectively, promote and inhibit both Rac and integrin activation in NCCs. Authors conclude that a global competition between Sdf1 and Sema3s for the control of NCC adhesion to fibronectin at the single cell level would be translated into directional migration due to the non-homogenous distribution of fibronectin itself in a complex 3D environment. Albeit potentially interesting, the hypothesis of Authors remains theoretical since it is not supported by a robust mechanistic data. In particular, the in vitro models are weakly connected to each other and do not recapitulate the complex 3D environment in which cephalic NCCs face in vivo. Furthermore, key controls are lacking in the in vivo experiments.

1. Neither in legend of Fig.1 nor in the "Materials and Methods" section, a description of how "controls" (shown in Fig. 1b-e) were treated is available. As the authors are no doubt aware, many morpholino off target effects have been shown. Authors have not even mentioned the standard 5 base pair mismatch control, which however it is no longer the best control. Authors should at least provide a rescue by sequential injection of morpholino and Sema3A or Sema3F mRNA. In any case, knocking out Sema3A and Sema3F, e.g. by CRISPR/Cas9, would be much more clean and reliable.

2. The series of in vitro experiments that the Authors put together to support their hypothesis of how the interplay among Sdf1, Sema3A, and Sema3F controls the generation and migration of cephalic NCCs in vivo is not convincing. This series of experiments is too far from the in vivo settings and does not faithfully mimic the complex in vivo environment. Transplantation of GFP-expressing NCCs coupled with gene knockdown and fluorescent microscopy over time (Borchers et al., 2000; Cousin, 2018) would be the ideal setting to firmly and directly validate in vivo the hypothesis of the Authors.

3. Several established aspects of the biology of Semaphorins are either not properly mentioned or overlooked:

A. On page 5, Authors state: "Ligand/receptor specificity is low since Nrp1 and Nrp2 can both act as co-receptors for either Sema3A or 3F". This is not exact. While Sema3F binds, albeit with significantly different affinities both Nrps, this does not apply to Sema3A that binds to Nrp1 only. In particular, the estimated dissociation constants (K_d) for Sema3F binding to Nrp1 and Nrp2 are 1.1 nM and 0.09 nM, respectively; the estimated K_d for Sema3A binding to Nrp1 is 1.15 nM. No specific binding of Sema3A to Nrp2 is instead detectable (Chen et al., 1997).

B. It has been known for 15 years that Semaphorins signal via the cytosolic split GTPase activating protein (GAP) domain of Plexins to inhibit integrin activation and cell adhesion to extracellular matrix proteins. The same applies to the fact that providing Semaphorins on living cells results in the dismantling of focal adhesion sites. For reviews, see: Kruger et al., 2005; Neufeld et al., 2007; Worzfeld and Offermanns, 2014. These notions should be taken into account and cited where

appropriate.

C. Inhibition of Rac1 activation that Authors observe in FRET upon treatment with Sema3A is likely an indirect effect derived from the inhibition of integrins, which in turn are known to support Rac1 GTP loading (Del Pozo and Schwartz, 2007). Indeed, upon activation by Semaphorins, Sema3A and Sema3F included (Bos and Pannekoek, 2012; Okada et al., 2015; Wang et al., 2012; Wang et al., 2013), the most direct enzymatic effect of the GAP domain of Plexins is to impair the GTP-loading of Rap1 small GTPase, which in turn is a well-known inducer of integrin activation (Calderwood et al., 2013; Zhu et al., 2017). Furthermore, Sema3A has been found to acutely promote, e.g. via Rac1 guanine nucleotide exchange factor FARP2, the GTP-loading of Rac1 that is required for Sema3A biological activity (Fournier et al., 2000; Jin and Strittmatter, 1997; Toyofuku et al., 2005; Västrik et al., 1999). The binding of small GTPases, such as Rac1 and Rnd1, to the Rho GTPase binding domain (RBD), which lies between the two halves of the split GAP cytodomain, fully unleashes the enzymatic activity of Plexins (Siebold and Jones, 2013). In sum, Rac1 activation is not an appropriate and direct readout of the mechanics of Sema3A/Plexin signaling. In this regard, based on the current knowledge, Rap1 would be the most reliable target to be investigated.

4. Nature Communications requests that Authors avoid "data not shown" statements and include data necessary to evaluate the claims of the paper. The same applies to the Supplementary Information.

REFERENCES

- Borchers, A., H.H. Epperlein, and D. Wedlich. 2000. An assay system to study migratory behavior of cranial neural crest cells in *Xenopus*. *Dev Genes Evol.* 210:217-222.
- Bos, J.L., and W.J. Pannekoek. 2012. Semaphorin signaling meets rap. *Sci Signal.* 5:pe6.
- Calderwood, D.A., I.D. Campbell, and D.R. Critchley. 2013. Talins and kindlins: partners in integrin-mediated adhesion. *Nat Rev Mol Cell Biol.* 14:503-517.
- Chen, H., A. Chedotal, Z. He, C.S. Goodman, and M. Tessier-Lavigne. 1997. Neuropilin-2, a novel member of the neuropilin family, is a high affinity receptor for the semaphorins Sema E and Sema IV but not Sema III. *Neuron.* 19:547-559.
- Cousin, H. 2018. Cranial Neural Crest Transplants. *Cold Spring Harb Protoc.* 2018:pdb.prot097402.
- Del Pozo, M.A., and M.A. Schwartz. 2007. Rac, membrane heterogeneity, caveolin and regulation of growth by integrins. *Trends Cell Biol.* 17:246-250.
- Fournier, A.E., F. Nakamura, S. Kawamoto, Y. Goshima, R.G. Kalb, and S.M. Strittmatter. 2000. Semaphorin3A Enhances Endocytosis at Sites of Receptor-F-actin Colocalization during Growth Cone Collapse. *J. Cell Biol.* 149:411-422.
- Jin, Z., and S.M. Strittmatter. 1997. Rac1 mediates collapsin-1-induced growth cone collapse. *J Neurosci.* 17:6256-6263.
- Kruger, R.P., J. Aurandt, and K.L. Guan. 2005. Semaphorins command cells to move. *Nat Rev Mol Cell Biol.* 6:789-800.
- Neufeld, G., T. Lange, A. Varshavsky, and O. Kessler. 2007. Semaphorin signaling in vascular and tumor biology. *Adv Exp Med Biol.* 600:118-131.
- Okada, T., S. Sinha, I. Esposito, G. Schiavon, M.A. López-Lago, W. Su, C.A. Pratilas, C. Abele, J.M. Hernandez, M. Ohara, M. Okada, A. Viale, A. Heguy, N.D. Socci, A. Sapino, V.E. Seshan, S. Long, G. Inghirami, N. Rosen, and F.G. Giancotti. 2015. The Rho GTPase Rnd1 suppresses mammary tumorigenesis and EMT by restraining Ras-MAPK signalling. *Nat Cell Biol.* 17:81-94.
- Siebold, C., and E.Y. Jones. 2013. Structural insights into semaphorins and their receptors. *Semin Cell Dev Biol.* 24:139-145.
- Toyofuku, T., J. Yoshida, T. Sugimoto, H. Zhang, A. Kumanogoh, M. Hori, and H. Kikutani. 2005. FARP2 triggers signals for Sema3A-mediated axonal repulsion. *Nat Neurosci.* 8:1712-1719.
- Västrik, I., B.J. Eickholt, F.S. Walsh, A. Ridley, and P. Doherty. 1999. Sema3A-induced growth-

cone collapse is mediated by Rac1 amino acids 17-32. *Curr Biol.* 9:991-998.

Wang, Y., H. He, N. Srivastava, S. Vikarunnessa, Y.B. Chen, J. Jiang, C.W. Cowan, and X. Zhang. 2012. Plexins are GTPase-activating proteins for Rap and are activated by induced dimerization. *Sci Signal.* 5:ra6.

Wang, Y., H.G. Pascoe, C.A. Brautigam, H. He, and X. Zhang. 2013. Structural basis for activation and non-canonical catalysis of the Rap GTPase activating protein domain of plexin. *Elife.* 2:e01279.

Worzfeld, T., and S. Offermanns. 2014. Semaphorins and plexins as therapeutic targets. *Nat Rev Drug Discov.* 13:603-621.

Zhu, L., J. Yang, T. Bromberger, A. Holly, F. Lu, H. Liu, K. Sun, S. Klapproth, J. Hirbawi, T.V. Byzova, E.F. Plow, M. Moser, and J. Qin. 2017. Structure of Rap1b bound to talin reveals a pathway for triggering integrin activation. *Nat Commun.* 8:1744.

Reviewer #2 (Remarks to the Author):

The authors examined the migration of neural crest (NC) cells in complex environment existing multiple guidance cues. They showed that knockdown of Sema3A and/or Sema3F disrupted the migration of NC cells (Fig. 1b-e). They found that Sdf1 overrode the inhibitory effect of Sema3A on NC cell migration in stripe assay (Supplementary fig 3 and movie 5). Knockdown of Sdf1 receptor Cxcr4 strongly suppressed the migration of NC cells but pre-activation of beta1-integrin with Mn⁺⁺ rescued the migration of the Cxcr4-knockdowned cells (Fig. 7c-e). Using primary cultured NC explants and/or dissociated cells, they showed that Sdf1 enhanced the motility of NC cells through the activation Rac1 (Fig. 4) and canceled the inhibitory effect of Sema3A including attenuation of cell dispersion (Fig. 2) and of focal adhesion on fibronectin (Fig. 3). In addition, activation of integrin function with Mn⁺⁺ also canceled the inhibitory effect of Sema3A (Fig. 5). This analysis provides novel information about NC migration. However, it lacks several essential evaluations including in vivo analysis of Rac1 activation during the migration. Furthermore, most of their in vitro experiments (Figs. 2 to 6) examined the cellular morphological response and/or random movement of NC cells. Considering the ultimate goal of this study, the assessment of NC directional motility and its molecular mechanism has to be experimentally designed.

Major comments

1) Rac1 activation and directional cell migration.

As extracellular stimulation transiently alters intracellular signaling, time course of Rac1 activation/inactivation should be provided in Fig. 4. The "override" effect of Sdf1 on Sema3A-induced Rac1 suppression should also be examined. Authors should simultaneously evaluate directional movement and subcellular Rac1 activity of NC cells after the local stimulation with Sdf1 or Sema3A (coated beads or stripes). In addition, in vivo Rac1 activity in NC cells may be visualized by Rac1 FRET or by labeling Rac1-GTP with GST-fused Pak1-BD probe.

The authors have shown that Sdf1 activates Rac1 in primary cultured NC cells with FRET-system while Sema3A inactivates the GTPase (Fig. 4). However, various studies have shown that Sema3A-Plexin-A cascade activates Rac1 in neurons (for example Toyofuku et al., *Nat Neurosci* 2005). Therefore, authors should discuss the difference between NC cell migration and neuronal response as well as the molecular mechanism of Rac1 regulation by Sdf1/Sema3A in NC cell migration with appropriate references.

2) Evaluation of fibronectin-integrin mediated NC migration

Pre-activation of integrin with Mn⁺⁺ in Cxcr4 knockdowned NC explants rescued the migration (Fig. 7d). Authors suggest that predominant expression of fibronectin in ventro-lateral regions may guide the NC-migration in the absence of Sdf1 navigation (Discussion, first paragraph). However, Figs. 7a and 7b indicate that fibronectin also expresses in dorsal region such as the boundary of neural plate and of notochord. Thus, the directional, ventro-lateral migration of NC cells may be regulated by additional molecule(s). Several scenarios including the involvement of dorsal

expression of *Sema3A/3F* may be examined to clarify the molecular mechanism of integrin-mediated ventro-lateral NC migration.

Other points

Figure presentation should be re-considered. Supplementary Fig. 3 should be presented as one of main figures because it represents obvious effect of *Sdf1* on NC migration in the presence of *Sema3A*. As Figure 6 contains less important information, this may be placed in supplementary figures. In figure 7a, c, the labels Fibronectin/DAPI may be indicated by the immunostained colors (Green and Red).

The authors propose a hypothesis that local-biased distribution of guidance cues directs the migration rather than global topographic and/or gradient expression of those cues. However, as several studies have already proved the importance of topographic and gradient expression such as retinotectal axonal projection, this reviewer feels that this study rather represents the robustness of NC migration with the cooperation of multiple guidance cues including *Sdf1*, *Sema3A*, and fibronectin. This point may also be discussed in text.

Reviewer #3 (Remarks to the Author):

The manuscript by Bajanca et al. "In vivo topology converts competition for cell-matrix adhesion into directional migration" aims to analyze how migrating neural crest (NC) cells integrate input from activating and repelling factors as well as the local environment. The authors show that the chemokine *Sdf1* can counterbalance the repellent function of *Sema3a*, which functions as a negative guidance cue for NC cells: increasing concentrations of *Sema3a* inhibit cell dispersion of explanted *Xenopus* cranial NC cells and this effect can be reversed by addition of *Sdf1*. Similar effects are observed for other cellular parameters like cell size, protrusion formation, and focal adhesions. This is likely based on opposing effects on *Rac1* activity as FRET reporter assays indicate an increase of *Rac1* activity in NC cells in presence of *Sdf1* and a decrease in presence of *Sema3a*. Consistently, *Sema3a* treatment of NC cells can be rescued using a photoactivatable *Rac1* GEF (*Tiam1*).

Surprisingly, the authors find that the antagonistic effect of *Sema3a* and *Sdf1* seems to be matrix-dependent and there seems to be a species-dependent preference for matrix. *Sdf1/Sema3a* competition apparently requires a Fibronectin matrix in *Xenopus*, while mouse O9-1 NC cells show a similar competition on Matrigel, but not Fibronectin. Finally, the authors show that *Xenopus* NC cells, which are lacking the *Cxcr4* receptor – and therefore also the ability to respond to *Sdf1* – can migrate in vivo if integrin signaling has been ectopically activated. This suggests that the local environment/the extracellular matrix – may be sufficient to direct migration.

This study presents significant novel findings how migrating NC cells integrate information and also challenges the view of long-range guidance cues in NC migration. Thus, these data have the potential to be of major interest for authors interested in directed cell migration in the developmental or clinical context.

However, there are some points that should be addressed:

1. There is not sufficient information on the experimental methods as well as the data analysis (there is information on the statistical analysis). Although the authors use methods that have been previously published, these are only rarely cited. At least a brief summary of critical procedures would be helpful for the readers. This concerns measurements of cell dispersion, cell size, cell area, protrusion/focal adhesion formation, aspect ratio, and circularity in in vitro experiments as well as details on the *Xenopus* injection procedure (stage, injection site, concentration).

2. It is surprising that the Sdf1/Sema3a competition requires a Fibronectin matrix in *Xenopus*, but Matrigel for mouse O9 cells. How do the authors explain this on a molecular level? Are they suggesting that other ligands like VEGF replace Sdf1 in other species?

3. The *Xenopus* transplantation experiments seem to suggest, that a biased Fibronectin matrix is more important than Sdf1 signaling. What does this mean for the *in vivo* relevance of the Sdf1/Sema3a competition? Where and when would this antagonism be relevant for *in vivo* NC migration? Furthermore, how does this result fit in with previous publications of the authors showing a requirement for Sdf1 in directional migration (Theveneau et. al., *Developmental Cell*, 2010; Theveneau et. al., *Nature Cell Biology*, 2013)? Sdf1 MO injection inhibits cranial NC migration in *Xenopus*. Can migration also be restored if these embryos are treated with Mn²⁺?

4. The loss of function data presented for Sema3a and Sema3f (Fig. 1b) is inconclusive. First, MO controls (mismatch, alternative/splice MOs, rescue, ...), number of experiments and statistical analysis are missing. If rescue experiments are not possible, this should be indicated. Second, the phenotype may indicate a lack of confinement, however, it may simply result from a general inhibition of migration. For example NC cells in Sdf1-MO or dnCxcr4 MO injected embryos (Theveneau et. al., *Developmental Cell*, 2010) also appear fused. Consistently, with an inhibition in migration, the authors also indicate shorter NC streams (Fig. 1b, this manuscript). However, how can the loss of function of a repellent guidance cue lead to an inhibition in migration?

5. Fig. 6: The cells at high magnification seem not always to be representative of the cells shown at low magnification. Low magnification shows quite a large number of circular cells for the sema3A+Mn treated cells, while the cells at higher magnification appear polar. For comparison it would be better to show all conditions on Matrigel AND Fibronectin in Fig. 6A; especially as aspect ratio and cell circularity was not addressed in the previous figures.

6. In addition to point 1: The authors show antagonistic effects on "cell dispersion", however, the graph in Fig. 2c shows "Explant area". How was the explant area determined? As the area that is covered by cells would not be particularly meaningful I assume the authors calculated the "Explant area" using Delaunay triangulations?

7. Fig. 3b,c: I assume that the cell protrusion area is included in the term "cell size". Why is Sdf1 leading to an increase in the protrusion area, but the total cell size is not affected compared to controls?

8. Fig. 1S (b,c): double *in situ* would be preferable to show co-expression. What does "Xslug/Twist" in Fig. 1c mean? Were both probes used simultaneously?

9. Looking at the expression pattern of Sema3a,f the cartoon in Fig. 1f,e should be revised to make it clear that Semas are not expressed in a solid block of cells. For example the majority of cranial NC migrate in a significant distance of Sema3a expressing cells. Thus, even taken into account that Sema3a is a secreted factor these NC cells are likely not exposed to Sema3a.

10. Fig. 3: The authors state that they transfected cells (I assume these are *Xenopus* NC cells and these were injected with the respective constructs?). This is also mentioned on multiple occasions in the text.

11. Fig. 2b: The explants incubated with the highest concentration of sema3A look like they are at an earlier stage at t0 compared to the other conditions.

12. The sentence "Sdf1 and Sema3A had antagonistic effects on FAs (Fig. 3d), reducing the total area occupied by FAs..." is misleading. Only Sema3a reduces the total area occupied by FAs.

13. In addition to point 1: Although the FRET method has been cited a brief outline for the calculation of the FRET efficiency would be helpful. In addition the color code for the FRET efficiency should be provided in Fig. 4b.

14. The labeling in Fig. 4f is unclear. Which conditions are +/- Sema?

15. Fig. 3a: The red at 0 and 8 minutes is difficult to distinguish.

16. Supplementary Fig. 6: Why does addition of Sdf1 in Fig. 6e lead to a significant decrease in speed. Line 4 of the figure legend: "shown in (a)" should be "shown in (d)".

Please find below detailed point by point answers to the comments.

Reviewers' comments:

Reviewer #1 (Remarks to the Author):

1. Neither in legend of Fig.1 nor in the “Materials and Methods” section, a description of how “controls” (shown in Fig. 1b-e) were treated is available. As the authors are no doubt aware, many morpholino off target effects have been shown. Authors have not even mentioned the standard 5 base pair mismatch control, which however it is no longer the best control. Authors should at least provide a rescue by sequential injection of morpholino and Sema3A or Sema3F mRNA. In any case, knocking out Sema3A and Sema3F, e.g. by CRISPR/Cas9, would be much more clean and reliable.

On Figure 1 panels of injected embryos were labelled as “control side” and “injected side”. To reduce ambiguity, we replaced “control side” by “uninjected side”. In the legend it is stated that controls are non-injected embryos. We also added data with a standard control MO.

The 5-mismatch MO is not an appropriate control for aquatic species developing at low temperature (we keep *Xenopus* embryos in incubators at 12.5°C, 14.5°C or at room temperature 18°C in our *Xenopus* room, 21°C in our microscope room). 5 mismatches MO are calibrated to reduce (not abolish) binding to the target sequence in cells cultured at 37°C. We have tested the 5-mismatch MOs for several target genes and their ability to mimic the effect of the specific MO is clearly temperature dependent (the KD is nearly as good as that of the specific MO when embryos are kept for 48h at 12 to 14.5°C). We no longer use these type of MO. We have discussed this issue with gene-tools and they are aware of this and no longer recommend 5-mismatch MO for cells that do not grow at 37°C.

When possible the rescue is indeed the best control. However, semaphorins are expressed in a wide range of tissues and there is no blastomere at either 16 or 32-cell stage in *Xenopus* that would allow us to recapitulate their expression by targeted injections. Thus, it is not possible to add back semaphorin while maintaining their normal distribution. It was already known that sema3A and 3F act as inhibitors of NC migration in fish (by Morpholino KD, (1)), mouse (KO, (2)) and chick (overexpression of dominant-negative forms of plexins, graft of beads soaked in sema3A/3F and in vitro stripe assays (2-4))). The fact that our Morpholinos reproduce the phenotypes described in these three other species is a very strong arguments to say that the phenotype is real and not due to an artefact and that sema3A/3F have a conserved role in NC migration.

Nonetheless, we completely agree with the reviewer that loss-of-function need to be confirmed by alternative methods whenever possible. To this end we tested knockdown by CRISPR/Cas9. We designed three gRNA targeting different regions of the gene for each sema and co-injected them as a cocktail with either normal Cas9 or the double point mutant deadCas9 to block expression of sema3A/3F. We checked the efficiency of the KD by in situ hybridization (sup figure 2). mRNA for sema3A and 3F are clearly reduced on the injected side. These experiments mimic the effect of the MOs. Interestingly, the toxicity of these injection of Cas9 plus 3 gRNA

was significantly lower than that of the typical MO injection (regardless of the target gene). We will now use systematically this method in parallel with MO injection in the lab. Details of the protocol has been added to the methods and the new data have been added to Figure 1.

2. The series of in vitro experiments that the Authors put together to support their hypothesis of how the interplay among Sdf1, Sema3A, and Sema3F controls the generation and migration of cephalic NCCs in vivo is not convincing. This series of experiments is too far from the in vivo settings and does not faithfully mimic the complex in vivo environment. Transplantation of GFP-expressing NCCs coupled with gene knockdown and fluorescent microscopy over time (Borchers et al., 2000; Cousin, 2018) would be the ideal setting to firmly and directly validate in vivo the hypothesis of the Authors.

We respectfully disagree with this reviewer that some of our experiments aimed at analyzing the generation of NC cells. None of our experiments were designed to study either induction of NC cell fate or their delamination from the neuroepithelium. We only performed experiments analyzing migration.

The in vitro experiments we have performed showed that Semas reduce dispersion of explants, reduce adhesion to the substrate, reduce cell's directionality and reduce Rac1 activity. All these effects can be efficiently rescued by Sdf1. None of these aspects can be studied in vivo in *Xenopus* and thus in vitro assays were the standards to perform these analyses. All experiments were run with appropriate controls and statistical analyses.

Please note that in vivo analyses too are limited in their interpretation. For instance, in a previous project we had performed overexpression and inhibition of N-cadherin expression. Both lead to the exact same phenotype by in situ hybridization (arrest of NC migration). Yet it is only by in vitro assays that the effect of these manipulations can be revealed. N-cadh overexpression blocks dispersion whereas N-cadh inhibition promotes cell dissociation and inhibits contact-inhibition of locomotion and thus impairs cell polarity and dispersion. Therefore, in one case arrest of migration is due to cells locked into a solid-like configuration while in the other case cells have lost their directionality (see Kuriyama, Theveneau et al 2014 for details). This indicates that while in vitro assays are indeed far removed from in vivo settings, they provide nonetheless essential information about what aspect at the cellular level is affected and thus allow for interpretation of the in vivo phenotypes. In the current manuscript, sema-KD lead to shorter streams which could indicate that sema3A/3F regulate positively NC migration. In vitro assays demonstrate that they do not. In vivo data only make sense in light of the cell and molecular biology experiments that were performed in vitro.

Yet, we agree that it would be nice to have a dynamic view of cell migration in sema-KD conditions. Thus, as suggested by the reviewer, we performed live imaging of control NC cells in embryos injected with control MO, sema3A-MO or gRNA against sema3A-MO together with active Cas9 or dead Cas9. These movies show that NC cells migrating in a sema-KD environment are motile but fail to reach ventral regions of the face as efficiently as cells migrating in control environment. There is a wide range of phenotypes including delay, fusion of streams and unusual patterns of migration which are in agreement with the phenotypes we have described by in situ hybridization. These data have been added as supplementary Movie 1 and in Figure 2. These phenotypes are in line with what has been described for the KD of versican

another inhibitor of NC migration (Szabo et al JCB 2016). This shows that inhibitors are important to prevent random wandering of cells and help overall directionality (for discussion see Bronner 2016 JCB).

In addition, we showed in vitro that integrin activation via Mn²⁺ was sufficient to rescue exposure to sema. Since Sdf1 activates adhesion to Fibronectin we tested whether Mn²⁺ could rescue the absence of Sdf1 in vivo and it was indeed sufficient to compensate for the lack of Sdf1. This experiment is a very strong argument to support our findings and **indicates that data obtained in vitro were transferable in vivo.**

Furthermore, following this reviewer's concern we decided that it was important to try to rescue the absence of Sdf1 signaling in vivo by an alternative method that proved to be efficient in vitro. For that we co-injected Cxcr4MO with CIBN-GFP or CIBN-GFP+photoactivatable Tiam1. We then grafted these cells into control embryos and performed time-lapse imaging by illuminating the embryos every 10 minutes under an epifluorescence microscope (470nm LED). This type of illumination is not targeted, the whole dish is exposed at every time step of the movie. Yet, in absence of directed illumination cells injected with Cxcr4MO and Tiam were able to migrate in a dorso-ventral fashion whereas cells that had received Cxcr4MO and CIBN alone were not able to do so. We added these data in Figure 9 and sup Movie 13. These in vivo results strengthen the message of our article by showing that the maintenance of a basal level of Rac activation is sufficient to trigger dorso-ventral migration in absence of Sdf1. This indicates that the topology of the head at these stages of development contains a sufficient amount of directional information for cells to preferentially migrate in that direction without the need of a precise chemotaxis gradient as long as cells are able to adhere to the substrate.

(Please note: Cxcr4MO was previously validated in Theveneau et al 2010 by rescue and by comparison with dom-negCxcr4, chemical inhibitor and comparison with Sdf1-MO, also validated by rescue in Theveneau et al 2010).

3. Several established aspects of the biology of Semaphorins are either not properly mentioned or overlooked:

A. On page 5, Authors state: "Ligand/receptor specificity is low since Nrp1 and Nrp2 can both act as co-receptors for either Sema3A or 3F". This is not exact. While Sema3F binds, albeit with significantly different affinities both Nrps, this does not apply to Sema3A that binds to Nrp1 only. In particular, the estimated dissociation constants (Kd) for Sema3F binding to Nrp1 and Nrp2 are 1.1 nM and 0.09 nM, respectively; the estimated Kd for Sema3A binding to Nrp1 is 1.15 nM. No specific binding of Sema3A to Nrp2 is instead detectable (Chen et al., 1997).

In the paper by Chen and colleagues (Chen et al, 1997, Neuron) the binding assay is performed with COS cells expressing Nrp1 or various forms of Nrp2 incubated with three different semaphorins coupled to Alkaline Phosphatase. They showed that in these cells Nrp2 does not interact with sema3A. Nasarre et al (2009) showed that Nrp2 can modulate Sema3A signaling during glioma cell migration (<https://www.ncbi.nlm.nih.gov/pmc/articles/PMC2802752/>). Since binding between Nrp2 and Sema3A was indeed controversial the authors directly tested it (see

the supp figure 1 in Nasarre et al 2009). We had cited this article in support of our claim. What these conflicting data suggest is that Nrp2 and Sema3A can (depending on the context) interact or not. However, since binding of sema3A to Nrp1 and 2 and sema3F to Nrp1 and 2 have been shown independently by different group of researchers, it is fair to say that ligand specificity is likely to be low. We added Chen et al in the reference list and, to make the statement clearer for the readers, we decided to rephrase this part. It now reads “Neuropilin 1 and 2 are expressed in cephalic NC cells (5). Nrp1 and Nrp2 can both act as co-receptors for either Sema3A and/or 3F *depending on the cell type studied* (6-8). Therefore, all NC cells could *theoretically* respond to both Sema3A/3F from the onset of migration.” We feel that this moderate version of the statement will raise awareness among the readers and prompt further scrutiny of the bibliography on this topic.

B. It has been known for 15 years that Semaphorins signal via the cytosolic split GTPase activating protein (GAP) domain of Plexins to inhibit integrin activation and cell adhesion to extracellular matrix proteins. The same applies to the fact that providing Semaphorins on living cells results in the dismantling of focal adhesion sites. For reviews, see: Kruger et al., 2005; Neufeld et al., 2007; Worzfeld and Offermanns, 2014. These notions should be taken into account and cited where appropriate.

We apologize if the reviewer felt that some works were not cited adequately. We have added the mentioned references to the revised version of the article.

C. Inhibition of Rac1 activation that Authors observe in FRET upon treatment with Sema3A is likely an indirect effect derived from the inhibition of integrins, which in turn are known to support Rac1 GTP loading (Del Pozo and Schwartz, 2007).

We agree with this statement that is why we had written “Rac1 is upstream of both actin polymerization and FAs” and “FA signalling can feed back into Rac1”. That is why we had tested direct modulation of Rac by Mn²⁺ treatments in our cells to confirm that the multiple entry points on Rac1 do exist in *Xenopus* NC cells. (see Figure 7 –former figure 5).

Indeed, upon activation by Semaphorins, Sema3A and Sema3F included (Bos and Pannekoek, 2012; Okada et al., 2015; Wang et al., 2012; Wang et al., 2013), the most direct enzymatic effect of the GAP domain of Plexins is to impair the GTP-loading of Rap1 small GTPase, which in turn is a well-known inducer of integrin activation (Calderwood et al., 2013; Zhu et al., 2017). Furthermore, Sema3A has been found to acutely promote, e.g. via Rac1 guanine nucleotide exchange factor FARP2, the GTP-loading of Rac1 that is required for Sema3A biological activity (Fournier et al., 2000; Jin and Strittmatter, 1997; Toyofuku et al., 2005; Västriik et al., 1999). The binding of small GTPases, such as Rac1 and Rnd1, to the Rho GTPase binding domain (RBD), which lies between the two halves of the split GAP cytodomain, fully unleashes the enzymatic activity of Plexins (Siebold and Jones, 2013). In sum, Rac1 activation is not an appropriate and direct readout of the mechanics of Sema3A/Plexin signaling. In this regard, based on the current knowledge, Rap1 would be the most reliable target to be investigated.

This reviewer is right that sema-3A can signal via Rac1 to regulate growth cone collapse. It is also true that Sema-3A can impair GTP-loading of Rap1. However, nothing is known about Rap1 in *Xenopus* NC cells. By contrast Rac1 and RhoA have been studied previously and their activities are linked with formation and collapse of protrusions. Sdf1 stabilizes protrusions via activation of Rac1 (Theveneau et al 2010) whereas Wnt-PCP and N-cadherin respectively activates RhoA signaling and inhibits Rac1 (Carmona-Fontaine Nature 2008, Theveneau Dev Cell 2010). Since sema3A and Sdf1 have opposite effect on NC cell adhesion and motility it made sense to monitor Rac1 activity upon treatments with either molecules and in combination.

The fact that we find results in NC cells that diverge from what has been found in neurons in culture (retinal ganglion, dorsal root ganglia, PC12 cell line induced to form neurites in vitro) does not change the results we obtained in NC cells and is in our opinion anecdotic. It is not unusual to find opposite effect on downstream effectors of a given pathway when looking at different cell types in different contexts. There are numerous feedbacks between GTPases and between small GTPases and other signaling pathways. Since the relationship between semaphorin signaling and Rac1 is unlikely to be direct (as this reviewer has pointed out) the observed effect on Rac1 will depend on the cellular context. It may very much depend on the integrin repertoire that is being expressed and on how this repertoire influences Rac levels and also on the type of guidance molecules growth cones are receiving at the time. Growth cones are very sensitive to modulation of Rac1 since activation and inhibiting Rac both lead to collapse. We added a paragraph to the discussion to address these discrepancies.

It may be interesting to look at Rap1 signaling too, but that is beyond the scope of this study.

4. Nature Communications requests that Authors avoid "data not shown" statements and include data necessary to evaluate the claims of the paper. The same applies to the Supplementary Information.

In the previous version of this paper, we used data not shown in three occasions. First, to refer to Nrp1 and 2 expression patterns. Our data were identical to that of Borchers and colleagues so we removed that claim and simply refer to their publication. Second, to mention Phalloidin staining in the description of former figure 3 (now figure 4). We added the phalloidin and dapi staining to figure 4. Third, we were referring to the PCR data confirming expression of Nrp1/2 and Cxcr4 in O9-1 cells, we added the PCR details to sup figure 8.

Here is the complete gel with the PCR results from two batches of O9-1 cells cDNA (A, B).

For the sup Figure, we edited it to remove lines with genes that we had monitored but that are not related to this story. (From left to right: MMP14, Cxcr4, Sox9, MMP28, Ets1, Cxcr7, Nrp1, Nrp2)

Reviewer #2 (Remarks to the Author):

Major comments

1) *Rac1* activation and directional cell migration.

As extracellular stimulation transiently alters intracellular signaling, time course of Rac1 activation/inactivation should be provided in Fig. 4.

We do not have access to actual Rac1 activation level in the PA-Tiam1 experiments. Expression of this GFP and mCherry tagged proteins is not compatible with the use of the CFP-YFP Rac1 FRET reporter. We calibrated the response to illumination at 488nm by looking at membrane relocalisation of Tiam1-mCherry (sup figure 7). Once we had these conditions working we applied them to our various experimental set-ups. We then analyzed membrane remodeling (formation of protrusions as a readout of Rac1 being activated) upon illumination.

Then, the time course of photoactivation is provided in the sup movie (green channel going ON/OFF). On the figure we have plotted the average cell area in OFF periods and ON periods during transient activation for the various conditions (Fig. 6f-g.) and the cell areas during constant activation in figure 6i-j. If the reviewer means that we did not provide details of any lag between exposure to light and cell's response, please note that cell protrusion form within 2 to 3 minute after green light has been turned on. We have added these details to the method section.

The “override” effect of Sdf1 on Sema3A-induced Rac1 suppression should also be examined.

We added FRET data in explants showing Rac1, RhoA and Cdc42 levels upon co-exposure to Sdf1 and Sema3A (Figure 6a). Numbers for Rac1 and RhoA are too low for statistical analysis but they show a clear trend (including a rescue of Rac1). In addition, we performed a new set of experiments using an antibody against Rac1-GTP (Figure 6c-e). Sdf1 rescues both the distribution and levels of Rac1-GTP upon exposure to sema3A.

Authors should simultaneously evaluate directional movement and subcellular Rac1 activity of NC cells after the local stimulation with Sdf1 or Sema3A (coated beads or stripes).

We feel that the proposed experiment with local activation of Sdf1 (beads) is not relevant to the current story since we show that precise/discrete localization of Sdf1 is dispensable in vivo. In addition, analysis of Rac1 in Sdf1 gradients has been performed in Theveneau et al 2010.

Here we show that inhibition of Sdf1 signaling via Cxcr4MO can be rescued in vivo by treatments with Mn²⁺ and via global activation of Rac1 (new data added to Figure 9). This means that, while NC cells can be seen chemotaxing to a local source of Sdf1 in vitro, our in vivo data strongly indicate that such gradient is not relevant in vivo.

Monitoring Rac1 FRET on sema stripes is also technically not possible. Stripes need to be stained to be detectable. However, for CFP-YFP FRET probes to be used one cannot use an additional color as it interferes with the FRET. Without knowing precisely where the sema+/sema-boundary is one cannot correlate local measurements of Rac1 with exposure to sema. In addition, cells extend protrusions on sema stripes for very brief periods of time before retracting the protrusions. So attempting such FRET experiments is also not possible on fixed cultures on which sema striped would be counterstained after FRET is acquired. Without the dynamics it is impossible to see at what step a cell was fixed. Before extending a lamellipodia on sema (high rac) or just after it has collapsed (low Rac). We had attempted such experiment early on in the project. It is simply impossible to be sure what we are looking at.

Instead, to try to rescue the absence of Sdf1 signaling in vivo by a global stimulation of Rac1, we co-injected Cxcr4MO with CIBN-GFP or CIBN-GFP+photoactivatable Tiam1. We then grafted these cells into control embryos and performed time-lapse imaging by illuminating the embryos every 10 minutes under an epifluorescence microscope. This type of illumination is not targeted, the whole dish is exposed at every time step of the movie. Yet, in absence of directed illumination cells injected with Cxcr4MO and Tiam were able to migrate in a dorso-ventral fashion whereas cells that had received Cxcr4MO and CIBN alone were not able to do so. We added these data in Figure 9 and sup Movie 13. These in vivo results strengthen the message of our article by showing that the maintenance of a basal level of global Rac activation is sufficient to trigger dorso-ventral migration in absence of Sdf1. This indicates that the topology of the head at these stages of development contains a sufficient amount of directional information for cells to preferentially migrate in that direction without the need of a precise chemotaxis gradient as long as cells are able to adhere to the substrate.

In addition, in vivo Rac1 activity in NC cells may be visualized by Rac1 FRET or by labeling Rac1-GTP with GST-fused Pak1-BD probe.

While we have tried to address the reviewer's request we could not find a successful approach due to limitations of the available tools and techniques.

The proposed GST-fused Pak1-BD probes are not specific to Rac1 but bind both Rac1 and Cdc42. In addition, active GTPases that are GTP loaded and interacting to effectors are not available for binding to the probe. Therefore, this tool, in addition to not being specific to Rac1, also binds to only a small fraction of GTP-loaded GTPases which does not allow for a precise assessment of Rac1 activity.

The original plasmid in which the Rac1 FRET probe is does not allow for mRNA synthesis. We had subcloned it in pCS2+ for in vitro RNA synthesis however injecting this RNA is toxic (may be due to higher level of expression than when using DNA?). We never managed to find the right balance between enough expression and toxicity. When we inject DNA we get mosaic expression because DNA is known to be trapped by nuclear proteins shuttling in the cytoplasm of *Xenopus* cells. This reduces dramatically the diffusion of DNA and thus, upon cell division, daughter cells inherit variable amounts of the plasmid. While this is not an issue in vitro as one can select only positive cells to look at, it is a problem in vivo when one needs to know where cells are within the NC stream. Cells that are the front facing a free space and cells completely surrounded by other NC cells are expected to have different Rac1 levels. Thus interpretation of the in vivo Rac1 activity is linked to the relative position of cells within the streams. To be able to do the FRET images one cannot add a counterstaining such as nuclear-mCherry or Fluorescein to visualize the whole NC population because it would interfere with the wavelengths required for FRET. Therefore, correlating Rac1 FRET levels with position in vivo is not possible in absence of counterstaining when Rac1 is mosaic. Performing multiple injections of the DNA reporter mitigates only partially the mosaicism but increases dramatically the toxicity. In summary, it is not technically possible to perform this experiment in vivo with reliable results.

We tried to use the anti-Rac1-GTP antibody on wholemount immunostaining or cryosection from gelatin-embedded *Xenopus* embryos. Despite it works nicely in vitro, we did not succeed in getting reliable reproducible staining in toto or on sections.

The authors have shown that Sdf1 activates Rac1 in primary cultured NC cells with FRET-system while Sema3A inactivates the GTPase (Fig. 4). However, various studies have shown that Sema3A-Plexin-A cascade activates Rac1 in neurons (for example Toyofuku et al., Nat Neurosci 2005). Therefore, authors should discuss the difference between NC cell migration and neuronal response as well as the molecular mechanism of Rac1 regulation by Sdf1/Sema3A in NC cell migration with appropriate references.

This is indeed interesting. We have added a paragraph to the discussion.

2) Evaluation of fibronectin-integrin mediated NC migration

Pre-activation of integrin with Mn⁺⁺ in Cxcr4 knockdowned NC explants rescued the migration (Fig. 7d). Authors suggest that predominant expression of fibronectin in ventrolateral regions may guide the NC-migration in the absence of Sdf1 navigation (Discussion,

first paragraph). However, Figs. 7a and 7b indicate that fibronectin also expresses in dorsal region such as the boundary of neural plate and of notochord. Thus, the directional, ventro-lateral migration of NC cells may be regulated by additional molecule(s). Several scenarios including the involvement of dorsal expression of Sema3A/3F may be examined to clarify the molecular mechanism of integrin-mediated ventro-lateral NC migration.

We think that there is a bit of confusion here. The neural plate-notochord boundary indeed contains Fibronectin but is located ventrally to the NC cells. NC cells are at the same level as the neural plate on the dorso-ventral axis, therefore, anything underneath the neural plate is also ventral to the NC, not dorsal to it.

We completely agree with the fact that additional signals could contribute to the bias. First we clarified the diagram summarizing the expression of the various molecules in Figure 1 and 10 to enhance the dorsal sema3A staining which is stronger than the ventro-lateral one. The notochord and the neural tube are epithelial structure and represent a physical barrier, and therefore contribute to making the dorso-medial region less permissive for migration. We made our statements clearer in the revised version of the paper when we comment on data shown in Figure 9 and in the discussion.

Other points

Figure presentation should be re-considered. Supplementary Fig. 3 should be presented as one of main figures because it represents obvious effect of Sdf1 on NC migration in the presence of Sema3A.

We moved sup fig. 3 to the main figures. It is now figure 5 in the revised version.

As Figure 6 contains less important information, this may be placed in supplementary figures.

We agree that findings presented in former figure 6 (now 7) may feel less important but since we managed to add all new data while staying in the maximum of 10 figures allowed for Nature Communications, we decided to keep this figure among the main figures. It shows a clear requirement for actin dynamics in the context of integrin-mediated rescue of exposure to sema3A which is an interesting point that may be otherwise lost in sup data.

In figure 7a, c, the labels Fibronectin/DAPI may be indicated by the immunostained colors (Green and Red).

We have introduced the reviewer's suggestion

The authors propose a hypothesis that local-biased distribution of guidance cues directs the migration rather than global topographic and/or gradient expression of those cues. However, as several studies have already proved the importance of topographic and gradient expression such as retinotectal axonal projection, this reviewer feels that this study rather represents the

robustness of NC migration with the cooperation of multiple guidance cues including Sdf1, Sema3A, and fibronectin. This point may also be discussed in text.

The case of retinotectal axonal projection is interesting in the sense that contrary to NC migration it occurs in an environment that, while still growing, conserves its overall shape. Under such condition one can easily conceptualize how gradients may be generated. However, NC cells migrate in the head while the mesoderm and the ectoderm are also undergoing intensive morphogenetic movements – formation of the brain vesicles, the eyes, the otic vesicles, aggregation of discrete placodes and formation of pharyngeal arches. In this context, robustness may be more easily achieved via other means than precise gradients that will be difficult to generate and maintain. We included a paragraph about the robustness of NC migration in the discussion.

Reviewer #3 (Remarks to the Author):

1. There is not sufficient information on the experimental methods as well as the data analysis (there is information on the statistical analysis). Although the authors use methods that have been previously published, these are only rarely cited. At least a brief summary of critical procedures would be helpful for the readers. This concerns measurements of cell dispersion, cell size, cell area, protrusion/focal adhesion formation, aspect ratio, and circularity in in vitro experiments as well as details on the Xenopus injection procedure (stage, injection site, concentration).

We apologize if the reviewer felt that experimental procedures were not properly described. We have added many details in the text and the Methods section following this reviewer recommendation.

2. It is surprising that the Sdf1/Sema3a competition requires a Fibronectin matrix in Xenopus, but Matrigel for mouse O9 cells. How do the authors explain this on a molecular level? Are they suggesting that other ligands like VEGF replace Sdf1 in other species?

Alfandari and colleagues have shown that Xenopus NC cells can adhere to numerous matrix molecules but can only migrate (displace their cell body efficiently) onto fibronectin. This showed a strict dependence of Xenopus NC cells to this matrix component and the authors had proposed it to be linked with the expression levels of different types of integrins. At neurula stage, the Xenopus embryo mostly express fibronectin and laminin. In mouse, like in chicken embryo, cephalic NC cells migrate on a richer matrix with several collagens, laminin, fibronectin, vitronectin. This is likely linked to a broader repertoire of integrins. Our data on O9-1 cells indicate that Sdf1 does not require the presence of Fibronectin to act since it can rescue exposure to sema3A on Matrigel. We don't know why the effect is not seen on Fibronectin with these cells. Identifying the various integrins involved in the differential response to Sdf1 and Sema3A in these two cell types is beyond the scope of this study. We felt that since Fibronectin has been proposed as a reservoir for Sdf1 it was interesting to show these data because they demonstrate that for signaling Sdf1 need not to be presented systematically via Fibronectin, the substrate requirement being likely species-specific (or cell-specific) and not Sdf-specific.

Would VEGF replace Sdf1 in chick or mouse? One cannot say since no experiments has been published testing this hypothesis directly. VEGF knockdown does not impair dorso-ventral migration of chick NC cells. However, it prevents NC cells from entering the branchial arches which are rich in semaphorins. One can suggest then that VEGF signaling might also contribute to the competition with semaphorins. VEGF is co-expressed with Sdf1 in chick and both molecules seem required. One big difference is that ectopic VEGF is not able to attract NC cells in between cephalic streams where semaphorins are expressed. A change of direction is noticed showing that VEGF can act as an attractant however it does not completely override semas like Sdf1. We have done similar experiments in Chick and Sdf1 is sufficient to attract NC cells in between cephalic streams in particular in the mesenchyme in front of rhombomere3 which is known to be rich in sema3A/3F.

3. The Xenopus transplantation experiments seem to suggest, that a biased Fibronectin matrix is more important than Sdf1 signaling. What does this mean for the in vivo relevance of the Sdf1/Sema3a competition? Where and when would this antagonism be relevant for in vivo NC migration?

We did not claim that the Fibronectin is more important than Sdf1. Sdf1 signaling is required for adhesion to Fibronectin via activation of Rac1. Thus, they are both needed. What we showed is that the precise positioning of Sdf1 is dispensable since global activation of cell-matrix adhesion via Mn²⁺ or Rac1 is sufficient to rescue Cxcr4MO.

Sema3A/3F and Sdf1 are broadly expressed at the onset of migration. Sdf1 is only expressed ventrally while semas are found both dorsally and ventrally. Thus the competition is relevant because cells are mostly exposed to Sdf1 along the latero-ventral ectoderm while Sdf1 is absent above the neural tube. This is one of the bias, the other ones being the distribution of fibronectin, a stronger expression of sema3A dorsally and the fact that the neural plate and the notochord are epithelial and may act as physical barriers. which we now discuss in more details in the revised discussion.

Recent data published by Barriga et al (Nature, 2018) when we submitted this manuscript show that the stiffness of the mesoderm on which the fibronectin is located needs to reach a critical threshold for NC migration to start. Our data show why NC go ventrally (Sdf/Fibronectin ventral bias, sema3A dorsal bias) whereas theirs show why NC start at a specific stage. Interestingly our data show that Fibronectin and Sdf1 are already in place prior to migration. In their study, Barriga et al experimentally increase the stiffness at a stage where NC are not yet migrating. This is sufficient to trigger an early migration in agreement with our description of Sdf1 expression pattern and Fibronectin distribution. Thus, NC cells need fibronectin, they need this fibronectin to be atop a tissue that is stiff enough and they need sdf1 to override semas to be able to form adhesions. We discuss these points in more details in the revised version of the paper.

Furthermore, how does this result fit in with previous publications of the authors showing a requirement for Sdf1 in directional migration (Theveneau et. al., Developmental Cell, 2010; Theveneau et. al., Nature Cell Biology, 2013)?

We had shown that while Sdf1 can promote directional migration in vitro it was unable to give directionality at the single cell level (unable to polarize Rac1 activity, Theveneau et al 2010). We have been always very careful to translate in vitro chemotaxis into an in vivo model of chemotaxis. Sdf1 expression pattern and its loss of function does not match the prediction for such chemotactic cues. We have discussed this point in numerous reviews and opinion articles over the years and these observations were the very basis of this present study.

The NC-placode behavior we described (Theveneau et al 2013) is in line with this submitted study. We had shown that Sdf1 was expressed directly adjacent to the NC territory making it unlikely that cells would use it as a directional cue to reach the distant ventral regions. What we showed then was that these two cell populations influence each other. There is a ventral bias because Sdf1⁺-placodes are located ventrally to NC cells and that can explain the directionality of the onset of migration. What we bring with this new study is that the ventral bias does not depend so much on Sdf1 distribution but on the fact that Sdf1 promotes adhesion to the substrate via activation of Rac1/Integrins since global activation of integrins and global activation of Rac1 are both sufficient to elicit dorso-ventral migration in absence of Cxcr4 signaling.

The chase'n'run behavior is nonetheless happening in vivo (we had published in vivo time-lapse movies showing both cell populations moving together) and is not a cell culture artefact. Since placodes are slower than NC cells, they eventually accumulate in between NC streams. They co-express Sdf1 and sema3F. So they provide three types of guidance: physical barrier (as epithelial structures aggregating in between NC streams), inhibition of adhesion via semaphorin and stimulation of motility via secretion of Sdf1. Sdf1 is smaller than semas so they may have different diffusion in vivo. A short-range semas vs a long-range sdf1 diffusion. During the work that led to Theveneau et al 2013, we had performed in vivo time-lapse movies that we never published in which we showed that once placodes have accumulated on the side of NC cells, NC cells turn to accumulate along placodes. They do a 90° turn suggesting that locally can sense Sdf1 being produced by placodes. Yet NC cells do not cross in between streams at this point because by then placodes have piled up as epithelial structures in between the streams. Epibranchial placodes are the source of sema3F (see the similarity between the in situ hybridization we showed on Figure 1 and placodal markers in Theveneau et al 2013). Therefore the chase-and-run behavior may also be modulated by semaphorins. We briefly discuss this point in the revised version.

Sdf1 MO injection inhibits cranial NC migration in Xenopus. Can migration also be restored if these embryos are treated with Mn2+?

We did not attempt this experiment. Sdf1 is needed for gastrulation and obtaining embryos in which gastrulation was OK but NC migration was impaired was very tricky during the project that led to Theveneau et al 2010. Injecting Cxcr4MO in the whole ectoderm (8-cell stage injection) is not an issue since NC cells derive from this tissue (but not gastrulating mesoderm). But injecting Sdf1MO targets the ectoderm from the roof of the blastocoele that will later give rise to the ectoderm of the neurula. Thus one cannot be precise enough to block NC migration independently of gastrulation in a systematic and reliable manner with Sdf1MO. That is why experiments in Theveneau et al 2010 relied mostly on Cxcr4MO instead. One can only guess that

since Mn²⁺ and Sdf1 are both sufficient to activate Rac1 and that Sdf1 does so via Cxcr4, Mn²⁺ would probably rescue lack of Sdf1 as it rescues lack of Cxcr4.

We interpret the idea behind this comment as to show with an alternative method that it is possible to compensate for lack of Cxcr4 signaling. Therefore, we rescued the absence of Sdf1 signaling in vivo by a global stimulation of Rac1. For that we co-injected Cxcr4MO with CIBN-GFP or CIBN-GFP+photoactivatable Tiam1. We then grafted these cells into control embryos and performed time-lapse imaging by illuminating the embryos every 10 minutes under an epifluorescence microscope. This type of illumination is not targeted, the whole dish is exposed at every time step of the movie. Yet, in absence of directed illumination cells injected with Cxcr4MO and Tiam were able to migrate in a dorso-ventral fashion whereas cells that had received Cxcr4MO and CIBN alone were not able to do so. We added these data in Figure 9 and sup Movie 13. These in vivo results strengthen the message of our article by showing that the maintenance of a basal level of Rac activation is sufficient to trigger dorso-ventral migration in absence of Sdf1. This indicates that the topology of the head at these stages of development contains a sufficient amount of directional information for cells to preferentially migrate in that direction without the need of a precise chemotaxis gradient as long as cells are able to adhere to the substrate.

4. The loss of function data presented for Sema3a and Sema3f (Fig. 1b) is inconclusive. First, MO controls (mismatch, alternative/splice MOs, rescue, ...), number of experiments and statistical analysis are missing. If rescue experiments are not possible, this should be indicated. Second, the phenotype may indicate a lack of confinement, however, it may simply result from a general inhibition of migration. For example NC cells in Sdf1-MO or dnCxcr4 MO injected embryos (Theveneau et al., Developmental Cell, 2010) also appear fused. Consistently, with an inhibition in migration, the authors also indicate shorter NC streams (Fig. 1b, this manuscript).

We apologize for the missing information of Figure 1. Regarding the statistics, we are sorry that we indeed forgot to add the number of embryos and independent experiments and statistics corresponding to figure 1b-e. We have now corrected this. Similarly, we had omitted the results with the standard control MO they have now been added back.

The standard 5-mismatch MO is not an appropriate control for aquatic species developing at low temperature (our *Xenopus* embryos are kept in incubators at 12.5°C, 14.5°C or at room temperature 18°C in our *Xenopus* room, 21°C in our microscope room). 5 mismatches are calibrated to reduce (not abolish) binding to the target sequence in cells cultured at 37°C. We have tested the 5-mismatch MOs for several target genes and their ability to mimic the effect of the specific MO is clearly temperature dependent (the KD is nearly as good as that of the specific MO at 12 and 14.5°C). We no longer use these type of MO. We have discussed this issue with gene-tools and they are aware of this and no longer recommend 5-mismatch MO for cells that do not grow at 37°C.

When possible, the rescue is indeed the best control. However, semaphorins are expressed in wide range of tissues and there is no blastomere at either 16 or 32-cell stage in *Xenopus* that would allow us to recapitulate their expression by targeted injections. Thus, it is not possible to add back semaphorin while maintaining their normal distribution.

It was already known that these two molecules act as inhibitors of NC migration in fish (Morpholino KD), mouse (KO) and chick (overexpression of dominant-negative forms of plexins and graft of beads soaked in sema3A/3F and in vitro stripe assays). The fact that our Morpholinos reproduce the phenotypes described in these three other species is a very strong argument to say that the phenotype is real and not due to an artefact and that sema3A/3F have a conserved role in NC migration.

Nonetheless, we completely agree with this reviewer that loss-of-function need to be confirmed by alternative methods whenever possible. To this end we tested knockdown by CRISPR/Cas9. We designed three gRNA targeting different regions of the gene for each sema and co-injected them as a cocktail with either normal Cas9 or the double point mutant deadCas9 to block expression of sema3A/3F. We checked the efficiency of the KD by in situ hybridization. mRNA for sema3A and 3F is clearly reduced on the injected side. These experiments mimic the effect of the MOs. Interestingly, the toxicity of these injection of Cas9 plus 3 gRNA was significantly lower than that of the typical MO injection (regardless of the target gene). We will now use systematically this method in parallel with MO injection in the lab. Details of the protocol has been added to the methods and the new data have been added to Figure 1. We also looked at NC migration in the various sema-KD conditions by time-lapse imaging (data added in Figure 2).

About the fusion of streams and the interpretation of the phenotype. The reviewer is right that in itself the interpretation is complex. Many treatments affect NC migration in vivo resulting in similar-looking embryos with shorter streams and partial fusion (LPAR2 inhibition, N-cadherin KD or overexpression, E-cadh over expression, Sdf1/Cxcr4-KD, inhibition of Wnt/PCP signalling etc e.g see Carmona-Fontaine 2008, Theveneau et al 2010; Theveneau et al 2013, Kuriyama 2014; Scarpa 2015). However, Cxcr4MO leads to a near complete arrest of migration while sema-KD leads to unprecise patterns with cells over the eye, located dorsally and in between streams. Migration still occurs under sema-KD. But this is still not sufficient to interpret. That is why in vitro assays to test the actual response of the cell to a given signal are done. Cxcr4MO cells adhere to fibronectin and are migratory. Cells exposed to sema detach from the substrate and show little capabilities to migrate. Only the cell biology experiments performed in addition to the in vivo loss-of-function allow us to formulate different hypothesis underpinning each phenotype.

For instance, overexpression and inhibition of N-cadherin expression lead to the exact same phenotype by in situ hybridization (arrest of NC migration). Yet it is only by in vitro assays that the effect of these manipulations can be revealed. N-cadh overexpression blocks dispersion whereas N-cadh inhibition promotes cell-cell dissociation and inhibits contact-inhibition of locomotion and thus impairs cell polarity. Therefore, in one case arrest of migration is due to cells locked into a solid-like configuration while in the other case cells have lost their directionality (see Kuriyama, Theveneau et al 2014 for details).

However, how can the loss of function of a repellent guidance cue lead to an inhibition in migration?

While at first this can be puzzling there is a simple explanation. In an inhibitor-free environment cells have many more options to migrate whereas in an environment containing inhibitors options are fewer and thus it has an overall positive feedback effect on directionality. In other words

inhibitors help directionality by reducing the probability of random wandering. This effect has been demonstrated in vivo and in vitro for Versican in *Xenopus* NC cells and tested in silico (see Szabo et al JCB 2016). This has been commented in J Cell Biol (see Bronner JCB 2016 for discussion around the idea that negative signals promote directionality by shaping migratory routes).

At the onset of migration semaphorins are located dorsally and ventrally to the NC while fibronectin and *sdf1* are only ventral, thus there is a bias towards ventral regions. If one removes semaphorin, it opens a possibility for dorsal migration (see ectopic cells located above the neural tube in figure 1e and 1g) reducing the pressure for ventral movement explaining the overall delay in reaching ventral sites. As NC cells migrate ventrally placodal cells progressively accumulate in discrete locations helping to shape the streams (see Theveneau et al Nat Cell Biol 2013). Since placodal cells express semaphorins, lack of sema most likely reduces the inhibitory effect of placodes accounting for the cells wandering in between streams. Placodes eventually aggregates into epithelial structures in between NC streams, at this point expression of *sema3F* in placodes may become dispensable for NC patterning.

As a follow-up from this project and from Theveneau et al 2013, it would be interesting to test whether sema produced from placodal cells play a direct role in the chase'n'run behavior and whether they influence the heterotypical contact-inhibition response we have described.

5.Fig. 6: The cells at high magnification seem not always to be representative of the cells shown at low magnification. Low magnification shows quite a large number of circular cells for the *sema3A*+Mn treated cells, while the cells at higher magnification appear polar.

The panel *sema3A*+Mn at low magnification contains 16 cells of which 6 are round (actin uniformly distributed as an uninterrupted circle). That represent a ratio of $6/16 = 0.375$. The high magnification panel contains 4 cells. The two cells on the left are bipolar. The cell on top is clearly unpolarized. The last cell is not round (seem to have two sides) but shows no significant difference in actin distribution that would allow leading and trailing edge to be identified. If we count this last cell as half-polarized that makes 1.5 cells over 4 that are round and that equals to a ratio 0.375. We feel that it accurately represents the proportions showed in the low magnification image. High magnification images bring more details and cells are sometimes more difficult to put in one category or the other. The zoom was carefully chosen to reflect the overall diversity of phenotypes. Please note that statistics on O9-1 cells involved thousands of cells (see legends of Figure 6 for details) it is not easy to find a field of view that would recapitulate such a large dataset.

For comparison it would be better to show all conditions on Matrigel AND Fibronectin in Fig. 6A; especially as aspect ratio and cell circularity was not addressed in the previous figures.

We agree with the reviewer. We added the data about circularity and aspect ratio in Figure 4 when we introduce cell shape measurements for the first in the paper. We refer to it when commenting the data on former Figure 6 (now 7).

6. In addition to point 1: The authors show antagonistic effects on “cell dispersion”, however, the graph in Fig. 2c shows “Explant area”. How was the explant area determined? As the area

that is covered by cells would not be particularly meaningful I assume the authors calculated the “Explant area” using Delaunay triangulations?

We measure explant area by drawing a line between all cells located at the periphery of a given explant over time. Thus, like Delaunay triangulation the area includes that of cells and the space in between them. For previous projects, we had compared this method with Delaunay triangulation and found no significant difference in terms of the dispersion observed. Please note that Delaunay triangulation (while fancy) has several caveats. Cells need to be stained either by expression of nuclear fluorescent protein or counterstained with Dapi. While we have done this many times both staining methods have problems. Expression levels of nuclear protein or dapi staining are often not homogeneous and many cells are excluded from the triangulation after automatic thresholding of the image, artificially increasing the overall dispersion. One way to avoid this is to do everything under confocal microscope for better imaging. This is not possible as a routine procedure. Plus, for long movies (several hours) the toxicity due to laser exposure for high quality imaging is not negligible and slow scans required for high quality actually imaging lead to fuzzy images since live cells are moving. Therefore, using confocal is only realistically possible when dealing with fixed explants for this kind of analysis which would not allow the overtime analysis we have performed in this study. Further, when working with treatments that block dispersion, non-dispersing cells are too compact to be segmented by automated image analysis. In these cases, dots have to be manually placed atop non-dissociating cells for triangulation to be made. It is impossible to guess where nuclei would be. Thus, one is forced to place dots at regular interval based on the average cell size in the whole area in which nuclei cannot be segmented. We felt that these caveats were too prone to artefacts and preferred to use a simpler method that can be used on any movie regardless of the imaging technic used and independently of the phenotype. It also bypasses the need for counterstaining or injection of cells which simplifies the procedure and ensures that all cells are included in the analysis. Since data are normalized to explant area at t0 (before dissociation starts) expansion rates are also corrected for variation in explant size.

7.Fig. 3b,c: I assume that the cell protrusion area is included in the term “cell size”. Why is Sdf1 leading to an increase in the protrusion area, but the total cell size is not affected compared to controls?

Yes, indeed we found that cell protrusions were bigger when cells are exposed to Sdf1 but that overall cell size is not affected. Protrusions are measured based on the area occupied by actin filaments in Phalloidin or life-act staining. In *Xenopus* cells lamellipodia are also devoid of vitelline platelets which provide an additional landmark in absence of other counterstaining. The basis of the lamellipodia (or lamella) where a change of fibers density (or orientation) is observed can also be used as a landmark to measure the area between this zone and the tip of the protrusion. The fact that cell size is overall not affected suggests that cells mostly change shape and mobilize their actin cytoskeleton differently but did not spread significantly more than controls. Controls NC cells on Fibronectin are already extremely spread. An increase of size can be seen when cells that were not very spread are treated with sdf1 (e.g. exposed to sema3A first and then to Sdf1). Similarly, and consistently with the Sdf1 results, please note that in Figure 6h activation of Rac/Tiam does not significantly increase cell area compared to control cells and that in figure 7d, Mn²⁺ does not increase cell area compared to control cells. Despite all 3 treatments

(sdf, Tiam, Mn²⁺) had clear effects on protrusions and focal adhesion and rescued cell size in sema treated cells.

8. Fig. 1S (b,c): double in situs would be preferable to show co-expression.

We completely agree. Unfortunately, we had previously attempted double staining with NBT/BCIP and BCIP alone to have purple and cyan blue staining but failed to get a sharp cyan blue staining as it diffuses. Attempt at performing fluorescent in situ were unsuccessful.

What does “Xslug/Twist” in Fig. 1c mean? Were both probes used simultaneously?

This comment refers to supp Figure 1 (not figure 1). It is our mistake. We had a previous version of this figure that contained a mix of early and late stages. Early stages were stained with Slug while late ones were stained with Twist. The submitted version only contains an early stage stained with Slug. We have corrected this mistake.

9. Looking at the expression pattern of Sema3a, the cartoon in Fig. 1f,e should be revised to make it clear that Semas are not expressed in a solid block of cells. For example the majority of cranial NC migrate in a significant distance of Sema3a expressing cells. Thus, even taken into account that Sema3a is a secreted factor these NC cells are likely not exposed to Sema3a.

The diagram in figure 1 depicts the embryo prior to the onset of migration and the colored areas are in agreement with the pictures showed in figure 1 and sup Fig1 for stage 17. Sema3A completely surrounds the NC domain. However, this comment made us aware of the fact that the diagram did not explicitly display the difference in expression levels between the dorsal domain of sema3A (which correspond to the anterior neural tube and is very high) and the ventral domain that has a weaker expression. We have corrected this in the new version. As for secretion of sema3A at late stages it is anyone's guess as there is currently no working antibody against Xenopus sema3A that could be used to assess the distribution of the protein in vivo. We would agree with this reviewer that it is unlikely that at late stages of migration NC cells located far from sources of sema3A would encounter the protein. Extracellular space being very restricted the probability of long-range diffusion of semaphorins is low.

10. Fig. 3: The authors state that they transfected cells (I assume these are Xenopus NC cells and these were injected with the respective constructs?). This is also mentioned on multiple occasions in the text.

Transfection refers to the deliberate introduction of nucleic acid into a cell. It is not a word that describes a specific technical procedure as transfection can be carried out by multiple means (e.g microinjection, lipofection, electroporation). We have clarified in the methods that Xenopus blastomeres were indeed microinjected but have left the word transfection in the text as we feel that it does not mislead the readers.

11.Fig. 2b: The explants incubated with the highest concentration of sema3A look like they are at an earlier stage at t0 compared to the other conditions.

That is correct they “look like” they are at an early stage. But, they are not at a younger stage, they only “look like” that because spreading and dispersion are impaired. Early NC cells are characterized by their inability to spread and disperse. To do that, they need to replace E-cadh by N-cadh and acquire contact-inhibition of locomotion (see Scarpa et al 2015). So any treatment that impairs the cadherin switch, increases the strength of cell-cell adhesion or prevent cell from adhering to the matrix will make the explant look like they are younger than they are. One good example is the inhibition of adhesion to fibronectin induced by adding N-cadherin into the substrate that we had published in Theveneau et al 2013 (see Figure 5 panels f, g and h). Explants exposed to N-cadh have problem to spread and indeed look like younger explants. Interestingly, N-cadh has a direct effect on focal adhesion size and distribution just like sema3A does and we had also shown previously that N-cadh represses Rac1 activity. Therefore it is not surprising that both treatments (sema3A and N-cadh) elicit a similar phenotype when mixed with Fibronectin.

All explants from Figure 2b are all coming from uninjected control embryos. NC cells were dissected at stage 17/18 when they are still adhering to one another but have delaminated from the neurectoderm layer. The only difference is the environment they encounter in the petri dish. Movies start after cells have been left on the bench to adhere and spread so that they can be carried to the microscope without detaching. So control cells or cells exposed to low concentrations of sema have time to start migrating a bit at the edge of the explant before time-lapse imaging starts. Cells exposed to high dose sema have problems to adhere and do not spread. In fact, it is very common that explants exposed to sema would detach from the fibronectin while being transported to the microscope. This “problem” of explants detaching was the very first clue that led us to study adhesion in more details.

In conclusion the fact that explants exposed to high dose of sema are “looking” younger is a direct consequence of the sema-dependent inhibition of adhesion.

12.The sentence “Sdf1 and Sema3A had antagonistic effects on FAs (Fig. 3d), reducing the total area occupied by FAs...” is misleading. Only Sema3a reduces the total area occupied by FAs.

Thank you for pointing this out. We have rephrased this part to avoid confusion.

13.In addition to point 1: Although the FRET method has been cited a brief outline for the calculation of the FRET efficiency would be helpful. In addition the color code for the FRET efficiency should be provided in Fig. 4b.

We have added the requested details in the methods.

14.The labeling in Fig. 4f is unclear. Which conditions are +/- Sema?

We changed the annotations below panels 4f and 4g to make clearer which conditions are with or without sema.

15.Fig. 3a: The red at 0 and 8 minutes is difficult to distinguish.

We agree. However, we have tried color scales and they were not more helpful to distinguish the first and last time points. We do not feel that this impairs the meaning of the figure. In addition, first and last time points can be seen in the associated sup movie.

16.Supplementary Fig. 6: Why does addition of Sdf1 in Fig. 6e lead to a significant decrease in speed.

Treatments that stabilize protrusions lower tumbling behavior which is observed when cells “hesitate” for several minutes before picking a new direction for a directional run. Speed and directionality that are measured during tracking includes phases of tumble when cells stop to pick a new direction. Cells are not immobile while tumbling. Instead, they show bursts of rapid movements in several directions before picking one for a stretch of directional movement. This often accounts for a slight increase of speed (despite not having a net displacement). Thus increasing directionality reduces the impact of tumbling on overall speed simply by reducing the frequency of tumbling events. An additional effect is the fact that stronger adhesion to the substrate is known to lead to slower (but more persistent) migration.

Line 4 of the figure legend: “shown in (a)” should be “shown in (d)”.

Indeed, thank you for pointing this out. We have corrected this mistake.

Selected references

1. H. H. Yu, C. B. Moens, Semaphorin signaling guides cranial neural crest cell migration in zebrafish. *Developmental biology* **280**, 373-385 (2005).
2. L. S. Gammill, C. Gonzalez, M. Bronner-Fraser, Neuropilin 2/semaphorin 3F signaling is essential for cranial neural crest migration and trigeminal ganglion condensation. *Developmental neurobiology* **67**, 47-56 (2007).
3. N. J. Osborne, J. Begbie, J. K. Chilton, H. Schmidt, B. J. Eickholt, Semaphorin/neuropilin signaling influences the positioning of migratory neural crest cells within the hindbrain region of the chick. *Developmental dynamics : an official publication of the American Association of Anatomists* **232**, 939-949 (2005).
4. B. J. Eickholt, S. L. Mackenzie, A. Graham, F. S. Walsh, P. Doherty, Evidence for collapsin-1 functioning in the control of neural crest migration in both trunk and hindbrain regions. *Development* **126**, 2181-2189 (1999).
5. U. Koestner, I. Shnitsar, K. Linnemannstons, A. L. Hufton, A. Borchers, Semaphorin and neuropilin expression during early morphogenesis of *Xenopus laevis*. *Developmental dynamics : an official publication of the American Association of Anatomists* **237**, 3853-3863 (2008).
6. C. Nasarre *et al.*, Neuropilin-2 acts as a modulator of Sema3A-dependent glioma cell migration. *Cell adhesion & migration* **3**, 383-389 (2009).

7. A. Sharma, J. Verhaagen, A. R. Harvey, Receptor complexes for each of the Class 3 Semaphorins. *Frontiers in cellular neuroscience* **6**, 28 (2012).
8. H. Chen, A. Chedotal, Z. He, C. S. Goodman, M. Tessier-Lavigne, Neuropilin-2, a novel member of the neuropilin family, is a high affinity receptor for the semaphorins Sema E and Sema IV but not Sema III. *Neuron* **19**, 547-559 (1997).

Reviewers' comments:

Reviewer #1 (Remarks to the Author):

Authors have painstakingly worked and addressed or convincingly discussed all my criticisms.

Guido Serini

Reviewer #2 (Remarks to the Author):

Comments for Bajanca F et al., revised manuscript of "In vivo topology converts competition for cell-matrix adhesion into directional migration."

This is a second review of this manuscript following revisions recommended by the first review. The authors have addressed most of the reviewer comments and requests for new data. In particular, regional Sema3A/3F knockout using CRISPR/Cas9 system (Fig. 1f-i), immunofluorescence of activated Rac1 in protrusion of NC cells (Fig. 6c), and rescue Cxcr4 loss-of-function by optogenetical Rac1 activation (Fig. 9f-h), these additional data strengthen the manuscript and make the conclusions more clear. This knowledge could have significant impact in our understanding of the regulatory mechanism of neural crest migration.

Minor comment.

As the main figures presenting in vitro assay lack directional migration of NC cells, supplementary Fig. 4 (Sema3A stripe & Sdf1 bead assay) may be presented as one of main figures.

Reviewer #3 (Remarks to the Author):

Concerning my previous comments on the manuscript most points have been successfully addressed. However, one critical point still remains and this concerns the in vivo loss of function data. The morpholino and especially the CRISPR/Cas experiments are not up to the standards in the field concerning quantity of embryos analyzed, tests for specificity and toxicity. For CRISPR/Cas, toxicity of single guide RNAs, Cas protein etc. should be analyzed; genomic mutations should be verified. Expression analysis by in situ hybridization is neither quantitative nor does it show specificity (the embryos shown in supplementary Fig. 2 b look very dismorph and the lack of Sema3F expression is not surprising – likely other markers would also be downregulated...).

Although the manuscript presents significant novel insight how neural crest cells integrate information in vitro, the in vivo relevance of these findings remains to be confirmed. Therefore, I cannot support publication of the manuscript in its current form.

Authors Response to Reviewers' comments:

Reviewer #3 (Remarks to the Author):

The authors have addressed my concerns and provided additional controls. However, I disagree with the statement that genomic analysis of CRISPR/Cas modified mosaic *Xenopus laevis* embryos is not informative. A convenient way to demonstrate the occurrence of genomic modifications would be to amplify genomic target sequences and analyze the mutation rate of the sequenced PCR products (see for example Hoff et al., JBC, 2018; Hsiao et al., bioRxiv, 2018). Nevertheless, overall the loss of function phenotypes are convincing and I therefore support publication of this manuscript.

We modified the result paragraph about CRISPR as follow (see sentence bold underlined): “To substantiate these data, we performed *sema3A/3F* knockdowns using CRISPR/Cas9 by co-injecting either an active Cas9 (*aCas9*) or a transcription-blocking catalytically inactive mutant form (dead Cas9, *dCas9*) with a cocktail of three guide RNAs against *sema3A* and/or *3F* (Fig. 1f-g)²⁸. Efficiency of CRISPR/Cas9 was assessed by *in situ* hybridization (Supplementary Figure 2). Importantly, injecting either active or dead Cas9 without gRNAs did not affect *Twist* expression or NC migration (Supplementary Figure 2). In addition, injecting the gRNAs against *sema3A* or *3F* without Cas9 did not affect *Twist* expression or NC migration (Supplementary Figure 2). **These internal controls for CRISPR specificity gave clear results. Thus, we did not further check for genomic modifications.** Overall, using CRISPR/Cas9, we observed phenotypes similar to those obtained with the MOs and all treatments led to significant increases in embryos with asymmetrical left-right NC migration (Fig. 1h) and embryos with NC cells in ectopic locations (Fig. 1i).”

We added the following statement in the methods, at the end of the CRISPR/Cas9 paragraph “All embryos were then analysed by *in situ* hybridization or used for grafting experiments. We did not keep some embryos to assess the occurrence of genomic modifications.”